# NEURAL TANGENT KERNELS FOR AXIS-ALIGNED TREE ENSEMBLES

## ABSTRACT

While *axis-aligned rules* are known to induce an important inductive bias in machine learning models such as typical hard decision tree ensembles, theoretical understanding of the learning behavior is largely unrevealed due to the discrete nature of rules. To address this issue, we impose the axis-aligned constraint on *soft trees*, which relax the splitting process of decision trees and are trained using a gradient method, and present their *Neural Tangent Kernel* (NTK) that enables us to analytically describe the training behavior. We study two cases: imposing the axis-aligned constraint throughout the entire training process, or only at the initial state. Moreover, we extend the NTK framework to handle various tree architectures simultaneously, and prove that any axis-aligned non-oblivious tree ensemble can be transformed into an axis-aligned oblivious tree ensemble with the same NTK. One can search for suitable tree architecture via Multiple Kernel Learning (MKL), and our numerical experiments show a variety of suitable features depending on the type of constraints, which supports not only the theoretical but also the practical impact of the axis-aligned constraint in tree ensemble learning.

## 1 INTRODUCTION

One of the most practical machine learning techniques used in real-world applications is *ensemble learning*. It combines the outputs of multiple predictors, often referred to as weak learners, to obtain reliable results for complex prediction problems. A hard decision tree is commonly used as a weak learner. Its inductive bias caused by the *axis-aligned splitting* of a feature space, which Figure 1(a) shows, is considered to be important. For example, Grinsztajn et al. (2022) experimentally demonstrated that the presence or absence of rotational invariance due to the axis-aligned constraint has a significant impact on generalization performance, especially for tabular datasets.

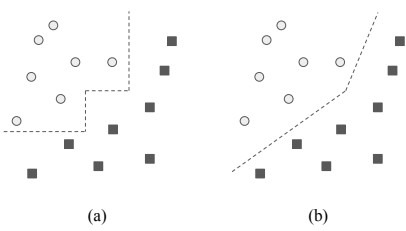

Figure 1: Splitting strategies. (a): Axis-aligned splitting, (b): Oblique splitting.

However, although a number of machine learning models have been proposed that are aware of axis-aligned partitioning (Chang et al., 2022; Humbird et al., 2019), it has not been theoretically clear what properties emerge when the axis-aligned constraints are imposed.

In this paper, we consider ensemble learning of *soft trees* to realize a differentiable analysis of axis-aligned splitting. A soft tree is a machine learning model that continuously relaxes the splitting process in a decision tree and is trained using a gradient method. There are various reasons why formulating trees in a soft manner is beneficial. Soft tree ensemble models are recognized for their high empirical performance (Kontschieder et al., 2015; Popov et al., 2020; Hazimeh et al., 2020). In addition, unlike hard decision trees, soft tree models can be updated sequentially (Ke et al., 2019) and trained in conjunction with pre-training (Arik & Pfister, 2021), resulting in desirable traits for continuous service deployment in real-world settings. Soft trees are also implemented in well-known open-source software such as PyTorch Tabular (Joseph, 2021), and its practical application is thriving.

Recently, there has been progress in the theoretical analysis of soft tree ensembles (Kanoh & Sugiyama, 2022; 2023) using the *Neural Tangent Kernel* (NTK) (Jacot et al., 2018). The NTK framework provides analytical descriptions of ensemble learning with infinitely many soft trees,

yielding several non-trivial properties such as the existence of tree architectures that have exactly the same training behavior even if they are non-isomorphic. However, the current NTK analysis assumes all the input features to be taken into account in each splitting process of soft trees, resulting in oblique splitting boundaries as shown in Figure 1(b). Therefore, it cannot directly incorporate the axis-aligned constraint in its current state.

In this paper, we extend the NTK concept to the axis-aligned soft tree ensembles to uncover the theoretical properties of the axis-aligned constraint. Our contributions can be summarized as follows:

- **Closed form solution of the NTK induced by axis-aligned tree ensembles.**

  We succeed in extending the prior work of Kanoh & Sugiyama (2022; 2023) to axis-aligned trees and successfully formulate a closed-form kernel induced by infinitely many axis-aligned tree ensembles. Using this kernel, we are able to analytically obtain the behavior of the training of axis-aligned trees (Theorem 2). We conducted a theoretical analysis on two cases: one in which the axis-aligned constraint is always imposed during training, and the other in which only the initial model is axis-aligned and training is conducted freely from there.

- **Deriving the NTK of ensembles of various tree architectures.**

  Prior studies (Kanoh & Sugiyama, 2022; 2023) were limited to analyzing the scenario of an infinite number of weak learners with identical architectures, which is not practical. This limitation is particularly unrealistic when analyzing axis-aligned trees, as a single feature is used at each split in axis-aligned trees (e.g., the second feature is always used at the first node in all trees, and the third feature is not used at any trees), which may cause a lack of representation power. We have successfully eliminated this unrealistic constraint and show that the NTK induced by a model that computes the sum of outputs from multiple sub-models is equal to the sum of the NTKs induced by each sub-model (Proposition 1). This proposition is applicable to not only tree ensembles but also any ensemble models such as Generalized Additive Models (GAM) (Hastie & Tibshirani, 1986).

- **Sufficiency of the oblivious tree for architecture search.**

  We show that any axis-aligned non-oblivious tree ensemble can be transformed into a set of axis-aligned oblivious tree ensembles that induces exactly the same NTK (Proposition 2). This proposition enables us to substantially reduce the number of potential tree architecture patterns.

- **Finding suitable tree architecture via Multiple Kernel Learning (MKL).**

  We employ MKL (Gönen & Alpaydın, 2011; Aiolli & Donini, 2015), which determines the weights of a linear combination of multiple kernels during training, to analyze the effect of the axis-aligned constraint in feature selection. The learned weights of the linear combination of NTKs induced by various tree architectures can be interpreted, using Proposition 1, as the proportion of the presence of each tree architecture. Our empirical experiments suggest that the suitable features vary depending on the type of training constraints.

## 2 PRELIMINARIES

### 2.1 SOFT TREE ENSEMBLES

We formulate regression using soft decision trees based on the literature (Kontschieder et al., 2015). Let $\boldsymbol{x} \in \mathbb{R}^{F \times N}$ be input data consisting of $N$ samples with $F$ features. Assume that there are $M$ soft decision trees and each tree has $\mathcal{N}$ splitting nodes and $\mathcal{L}$ leaf nodes. For each tree $m \in [M] = \{1, \ldots, M\}$, we denote trainable parameters of the $m$-th soft decision tree as $\boldsymbol{w}_m \in \mathbb{R}^{F \times \mathcal{N}}$ and $\boldsymbol{b}_m \in \mathbb{R}^{1 \times \mathcal{N}}$ for splitting nodes, which correspond to feature selection and splitting threshold in typical decision trees, and $\boldsymbol{\pi}_m \in \mathbb{R}^{1 \times \mathcal{L}}$ for leaf nodes.

Unlike typical hard decision trees, each leaf node $\ell \in [\mathcal{L}] = \{1, \ldots, \mathcal{L}\}$ in soft decision trees hold a value $\mu_{m,\ell}(\boldsymbol{x}_i, \boldsymbol{w}_m, \boldsymbol{b}_m) \in [0, 1]$ that represents the probability of input data reaching the leaf $\ell$:

$$\mu_{m,\ell}(\boldsymbol{x}_i, \boldsymbol{w}_m, \boldsymbol{b}_m) = \prod_{n=1}^{\mathcal{N}} \underbrace{\sigma(\boldsymbol{w}_{m,n}^\top \boldsymbol{x}_i + \beta b_{m,n})}_{\text{flow to the left}}{}^{\mathbb{1}_{\ell \swarrow n}} \underbrace{(1 - \sigma(\boldsymbol{w}_{m,n}^\top \boldsymbol{x}_i + \beta b_{m,n}))}_{\text{flow to the right}}{}^{\mathbb{1}_{n \searrow \ell}}, \quad (1)$$

where $\boldsymbol{w}_{m,n} \in \mathbb{R}^F$ is the $n$-th column vector of the matrix $\boldsymbol{w}_m$, $\beta \in \mathbb{R}^+$ is a hyperparameter that adjusts the influence of the splitting threshold, and $\mathbb{1}_{\ell \swarrow n}(\mathbb{1}_{n \searrow \ell})$ is an indicator function that returns 1 if the $\ell$-th leaf is on the left (right) side of a node $n$ and 0 otherwise.

Internal nodes use a decision function $\sigma : \mathbb{R} \to [0, 1]$ that resembles the sigmoid. This function is rotationally symmetric about the point $(0, 1/2)$ and satisfies the conditions: $\lim_{c \to \infty} \sigma(c) = 1$, $\lim_{c \to -\infty} \sigma(c) = 0$, and $\sigma(0) = 0.5$ for $c \in \mathbb{R}$. Examples of such functions include the two-class sparsemax function given as $\sigma(c) = \text{sparsemax}([\alpha c, 0])$ (Martins & Astudillo, 2016), and the two-class entmax function given as $\sigma(c) = \text{entmax}([\alpha c, 0])$ (Peters et al., 2019). In this paper, we mainly consider the scaled error function $\sigma(c) = \frac{1}{2} \text{erf}(\alpha c) + \frac{1}{2} = \frac{1}{2}(\frac{2}{\sqrt{\pi}} \int_0^{\alpha c} e^{-t^2} \, dt) + \frac{1}{2}$. As the scaling factor $\alpha \in \mathbb{R}^+$ (Frosst & Hinton, 2017) tends towards infinity, sigmoid-like decision functions become step functions that correspond to (hard) Boolean operation. Note that if we replace the right-flow term $(1 - \sigma(\boldsymbol{w}_{m,n}^\top \boldsymbol{x}_i + \beta b_{m,n}))$ with 0 in Equation 1, we obtain a rule set, which is represented by a linear graph architecture.

The function $f_m : \mathbb{R}^F \times \mathbb{R}^{F \times \mathcal{N}} \times \mathbb{R}^{1 \times \mathcal{N}} \times \mathbb{R}^{1 \times \mathcal{L}} \to \mathbb{R}$ that returns the prediction of the $m$-th tree is given by the weighted sum of leaf-specific parameters $\pi_{m,\ell}$, weighted by the probability that the input data $\boldsymbol{x}_i$ reaches each leaf:

$$f_m(\boldsymbol{x}_i, \boldsymbol{w}_m, \boldsymbol{b}_m, \boldsymbol{\pi}_m) = \sum_{\ell=1}^{\mathcal{L}} \pi_{m,\ell} \mu_{m,\ell}(\boldsymbol{x}_i, \boldsymbol{w}_m, \boldsymbol{b}_m). \tag{2}$$

Furthermore, the output of the ensemble model with $M$ trees for each input $\boldsymbol{x}_i$ is formulated as a function $f : \mathbb{R}^F \times \mathbb{R}^{M \times F \times \mathcal{N}} \times \mathbb{R}^{M \times 1 \times \mathcal{N}} \times \mathbb{R}^{M \times 1 \times \mathcal{L}} \to \mathbb{R}$ as follows:

$$f(\boldsymbol{x}_i, \boldsymbol{w}, \boldsymbol{b}, \boldsymbol{\pi}) = \frac{1}{\sqrt{M}} \sum_{m=1}^{M} f_m(\boldsymbol{x}_i, \boldsymbol{w}_m, \boldsymbol{b}_m, \boldsymbol{\pi}_m). \tag{3}$$

In general, parameters $\boldsymbol{w} = (\boldsymbol{w}_1, \ldots, \boldsymbol{w}_M)$, $\boldsymbol{b} = (\boldsymbol{b}_1, \ldots, \boldsymbol{b}_M)$, and $\boldsymbol{\pi} = (\boldsymbol{\pi}_1, \ldots, \boldsymbol{\pi}_M)$ are randomly initialized using independently and identically distributed normal distributions with mean 0 and variance 1 and updated using gradient descent. In this paper, the term "tree architecture" refers to both the graph topological structure of the tree and a parameter initialization method at each node. Even if the graph topology is identical, those architectures are considered to be distinct when different parameter initialization methods are adopted.

## 2.2 Neural Tangent Kernels

We introduce the NTK based on the gradient flow using training data $\boldsymbol{x} \in \mathbb{R}^{F \times N}$, the prediction target $\boldsymbol{y} \in \mathbb{R}^N$, trainable parameters $\boldsymbol{\theta}_\tau \in \mathbb{R}^P$ at time $\tau$, and an arbitrary model function $g(\boldsymbol{x}, \boldsymbol{\theta}_\tau) : \mathbb{R}^{F \times N} \times \mathbb{R}^P \to \mathbb{R}^N$. With the learning rate $\eta$ and the mean squared error loss function $L$, the gradient flow equation is given as

$$\frac{\partial \boldsymbol{\theta}_\tau}{\partial \tau} = -\eta \frac{\partial L(\boldsymbol{\theta}_\tau)}{\partial \boldsymbol{\theta}_\tau} = -\eta \frac{\partial g(\boldsymbol{x}, \boldsymbol{\theta}_\tau)}{\partial \boldsymbol{\theta}_\tau} (g(\boldsymbol{x}, \boldsymbol{\theta}_\tau) - \boldsymbol{y}). \tag{4}$$

Considering the formulation of the gradient flow in the function space using Equation 4, we obtain

$$\frac{\partial g(\boldsymbol{x}, \boldsymbol{\theta}_\tau)}{\partial \tau} = \frac{\partial g(\boldsymbol{x}, \boldsymbol{\theta}_\tau)}{\partial \boldsymbol{\theta}_\tau}^\top \frac{\partial \boldsymbol{\theta}_\tau}{\partial \tau} = -\eta \underbrace{\frac{\partial g(\boldsymbol{x}, \boldsymbol{\theta}_\tau)}{\partial \boldsymbol{\theta}_\tau}^\top \frac{\partial g(\boldsymbol{x}, \boldsymbol{\theta}_\tau)}{\partial \boldsymbol{\theta}_\tau}}_{\text{Neural Tangent Kernel:} \widehat{\boldsymbol{H}}_\tau(\boldsymbol{x}, \boldsymbol{x})} (g(\boldsymbol{x}, \boldsymbol{\theta}_\tau) - \boldsymbol{y}). \tag{5}$$

Here, we can see the NTK matrix $\widehat{\boldsymbol{H}}_\tau(\boldsymbol{x}, \boldsymbol{x})$. In general, the NTK can take two input datasets $\boldsymbol{x} \in \mathbb{R}^{F \times N}$ and $\boldsymbol{x}' \in \mathbb{R}^{F \times N'}$, resulting in $\widehat{\boldsymbol{H}}_\tau(\boldsymbol{x}, \boldsymbol{x}')$ with the shape of $\mathbb{R}^{N \times N'}$. The kernel value calculated for the two samples $\boldsymbol{x}_i$ and $\boldsymbol{x}_j$ is represented as $\widehat{\Theta}_\tau(\boldsymbol{x}_i, \boldsymbol{x}_j)$.

From Equation 5, if the NTK does not change during training, the formulation of the gradient flow in the function space becomes a simple ordinary differential equation, and it becomes possible to analytically calculate how the model's output changes during training. When the NTK is positive definite, it is known that the kernel does not change from its initial value during the gradient descent with an infinitesimal step size when considering an infinite number of soft binary trees (Lee et al., 2019; Kanoh & Sugiyama, 2022) under the formulation described in Section 2.1.

The NTK induced by a typical soft tree ensemble with infinitely many trees is known to be obtained in closed-form at initialization.

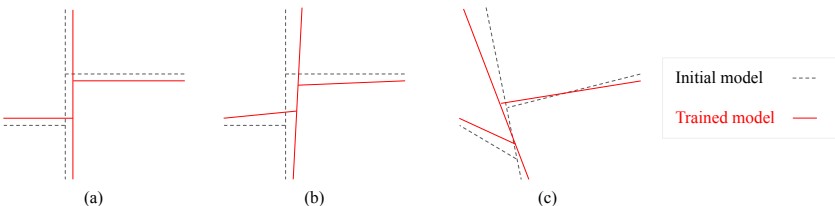

Figure 2: Splitting boundaries. (a): AAA, Always Axis-Aligned, during training, (b): AAI, Axis-Aligned at Initialization, but not during training, (c): Oblique splitting conducted by typical soft trees.

**Theorem 1** (Kanoh & Sugiyama (2023)). *Assume that all $M$ trees have the same tree architecture. Let $\mathcal{Q} : \mathbb{N} \to \mathbb{N} \cup \{0\}$ be a function that takes the depth as input and returns the number of leaves connected to internal nodes at that depth. For any given tree architecture, as the number of trees $M$ goes to infinity, the NTK probabilistically converges to the following deterministic limiting kernel:*

$$\Theta^{\text{Oblique}}(\boldsymbol{x}_i, \boldsymbol{x}_j) := \lim_{M \to \infty} \widehat{\Theta}_0^{\text{Oblique}}(\boldsymbol{x}_i, \boldsymbol{x}_j) = \sum_{d=1}^{D} \mathcal{Q}(d) \left( d \, \Sigma_{\{i,j\}} \mathcal{T}_{\{i,j\}}^{d-1} \dot{\mathcal{T}}_{\{i,j\}} + \mathcal{T}_{\{i,j\}}^{d} \right), \quad (6)$$

*where $\mathcal{T}_{\{i,j\}} = \mathbb{E}[\sigma(\boldsymbol{u}^\top \boldsymbol{x}_i + \beta v)\sigma(\boldsymbol{u}^\top \boldsymbol{x}_j + \beta v)]$, $\dot{\mathcal{T}}_{\{i,j\}} = \mathbb{E}[\dot{\sigma}(\boldsymbol{u}^\top \boldsymbol{x}_i + \beta v)\dot{\sigma}(\boldsymbol{u}^\top \boldsymbol{x}_j + \beta v)]$, and $\Sigma_{\{i,j\}} = \boldsymbol{x}_i^\top \boldsymbol{x}_j + \beta^2$. Here, the values of the vector $\boldsymbol{u} \in \mathbb{R}^F$ and the scalar $v \in \mathbb{R}$ are sampled from zero-mean i.i.d. Gaussians with unit variance. Furthermore, if the decision function $\sigma$ is the scaled error function, $\mathcal{T}_{\{i,j\}}$ and $\dot{\mathcal{T}}_{\{i,j\}}$ are obtained in closed-form as*

$$\mathcal{T}_{\{i,j\}} = \frac{1}{2\pi} \arcsin\left( \frac{\alpha^2(\boldsymbol{x}_i^\top \boldsymbol{x}_j + \beta^2)}{\sqrt{(\alpha^2(\boldsymbol{x}_i^\top \boldsymbol{x}_i + \beta^2) + 0.5)(\alpha^2(\boldsymbol{x}_j^\top \boldsymbol{x}_j + \beta^2) + 0.5)}} \right) + \frac{1}{4}, \quad (7)$$

$$\dot{\mathcal{T}}_{\{i,j\}} = \frac{\alpha^2}{\pi} \frac{1}{\sqrt{\left(1 + 2\alpha^2(\boldsymbol{x}_i^\top \boldsymbol{x}_i + \beta^2)\right)\left(1 + 2\alpha^2(\boldsymbol{x}_j^\top \boldsymbol{x}_j + \beta^2)\right) - 4\alpha^4(\boldsymbol{x}_i^\top \boldsymbol{x}_j + \beta^2)^2}}. \quad (8)$$

This kernel has rotational invariance with respect to input data.

## 3 THE THEORY OF THE NTK INDUCED BY AXIS-ALIGNED TREES

We first formulate the axis-aligned splitting (Section 3.1), and consider the NTK induced by axis-aligned soft tree ensembles, composed of weak learners with identical architectures, as assumed in Theorem 1 (Section 3.2). We then extend it to handle different tree architectures simultaneously (Section 3.3). Detailed proofs can be found in the Appendix A and B.

### 3.1 SETUP ON AXIS-ALIGNED SPLITTING

In Equation 1, input to the decision function $\sigma$ includes the inner product of $F$-dimensional vectors $\boldsymbol{w}_{m,n}$ and $\boldsymbol{x}_i$. Since $\boldsymbol{w}_{m,n}$ is typically initialized randomly, which is also assumed in Theorem 1, the splitting is generally oblique. Thus, Theorem 1 cannot directly treat axis-aligned tree ensembles.

To overcome this issue, we analyze the NTK when all the elements except one are set to be zero for every randomly initialized vector $\boldsymbol{w}_{m,n}$. This setting means that the corresponding features are eliminated from consideration of the splitting direction. This is technically not straightforward, as Gaussian random initialization is generally assumed in the existing NTK approaches. Parameters $\boldsymbol{b}$ and $\boldsymbol{\pi}$ are initialized with random values, as is commonly done.

We conduct a theoretical analysis of two cases: one where the parameters with zero initialization are not updated during training, as illustrated in Figure 2(a), and the other where they are updated during training in Figure 2(b). These two cases are referred to as AAA ("A"lways "A"xis-"A"ligned) and AAI ("A"xis-"A"ligned at "I"nitialization, but not during training) in this paper.

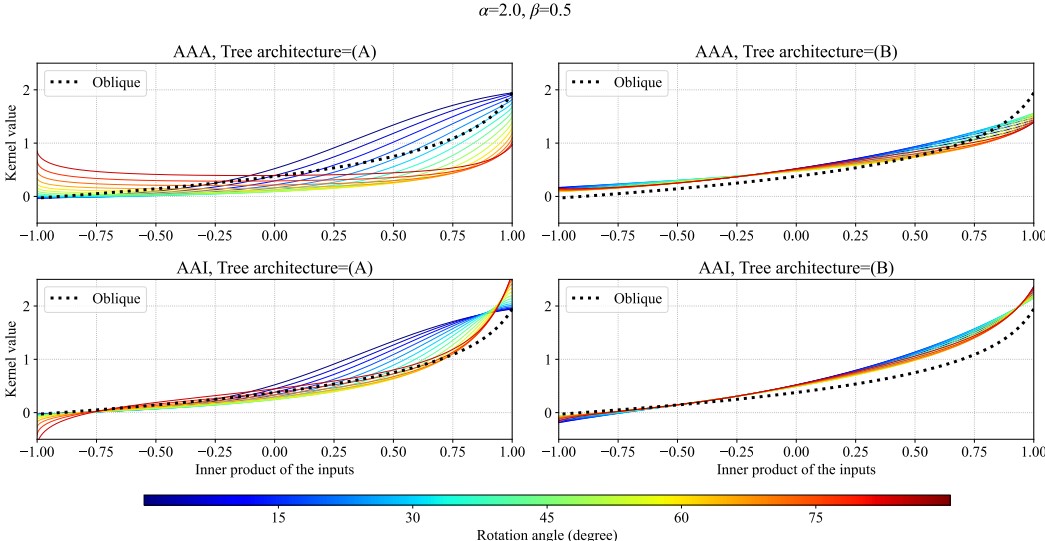

Figure 3: The rotation angle dependency of $\Theta^{\text{AxisAligned}}(\boldsymbol{x}_i, \boldsymbol{x}_j)$ with $\alpha = 2.0$ and $\beta = 0.5$. Different training procedures, AAA and AAI, are listed vertically, and two settings of tree architectures are listed horizontally. The dotted lines show the limiting NTK induced by typical oblique soft tree ensembles defined in Theorem 1, which is rotational invariant.

The place of the non-zero element of $\boldsymbol{w}_{m,n}$, which corresponds to the feature assigned to a node in a tree, needs to be predetermined before training. This is different from typical decision trees that search for features to be used for splitting during training. We address this issue by combining multiple kernels for multiple tree architectures via MKL in Section 4.2.

## 3.2 THE NTK INDUCED BY AXIS-ALIGNED SOFT TREES

For an input vector $\boldsymbol{x}_i$, let $x_{i,s} \in \mathbb{R}$ be the $s$-th component of $\boldsymbol{x}_i$. For both AAA and AAI conditions, at initialization, we derive the NTK induced by axis-aligned soft tree ensembles in a closed-form as the number of trees $M \to \infty$.

**Theorem 2.** *Assume that all $M$ trees have the same tree architecture. Let $\{a_1, \cdots, a_\ell, \cdots, a_\mathcal{L}\}$ denote the set of decomposed paths of the trees from the root to the leaves, and let $h(a_\ell) \subset \mathbb{N}$ be the set of feature indices used in splits of the input path $a_\ell$. For any tree architecture, as the number of trees $M$ goes to infinity, the NTK probabilistically converges to the following deterministic limiting kernel:*

$$\Theta^{\text{AxisAligned}}(\boldsymbol{x}_i, \boldsymbol{x}_j) \coloneqq \lim_{M \to \infty} \widehat{\Theta}_0^{\text{AxisAligned}}(\boldsymbol{x}_i, \boldsymbol{x}_j)$$

$$= \sum_{\ell=1}^{\mathcal{L}} \left( \sum_{s \in h(a_\ell)} \Sigma_{\{i,j\},s} \dot{\mathcal{T}}_{\{i,j\},s} \prod_{t \in h(a_\ell) \setminus \{s\}} \mathcal{T}_{\{i,j\},t} + \prod_{s \in h(a_\ell)} \mathcal{T}_{\{i,j\},s} \right), \quad (9)$$

*where $\mathcal{T}_{\{i,j\},s} = \mathbb{E}[\sigma(ux_{i,s} + \beta v)\sigma(ux_{j,s} + \beta v)]$ and $\dot{\mathcal{T}}_{\{i,j\},s} = \mathbb{E}[\dot{\sigma}(ux_{i,s} + \beta v)\dot{\sigma}(ux_{j,s} + \beta v)]$. Here, scalars $u, v \in \mathbb{R}$ are sampled from zero-mean i.i.d. Gaussians with unit variance. For $\Sigma_{\{i,j\},s}$, it is $x_{i,s} x_{j,s} + \beta^2$ when AAA is used, and $\boldsymbol{x}_i^\top \boldsymbol{x}_j + \beta^2$ when AAI is used. Furthermore, if the decision function is the scaled error function, $\mathcal{T}_{\{i,j\},s}$ and $\dot{\mathcal{T}}_{\{i,j\},s}$ are obtained in closed-form as*

$$\mathcal{T}_{\{i,j\},s} = \frac{1}{2\pi} \arcsin\left( \frac{\alpha^2(x_{i,s} x_{j,s} + \beta^2)}{\sqrt{(\alpha^2(x_{i,s}^2 + \beta^2) + 0.5)(\alpha^2(x_{j,s}^2 + \beta^2) + 0.5)}} \right) + \frac{1}{4}, \quad (10)$$

$$\dot{\mathcal{T}}_{\{i,j\},s} = \frac{\alpha^2}{\pi} \frac{1}{\sqrt{\left(1 + 2\alpha^2(x_{i,s}^2 + \beta^2)\right)\left(1 + 2\alpha^2(x_{j,s}^2 + \beta^2)\right) - 4\alpha^4(x_{i,s} x_{j,s} + \beta^2)^2}}. \quad (11)$$

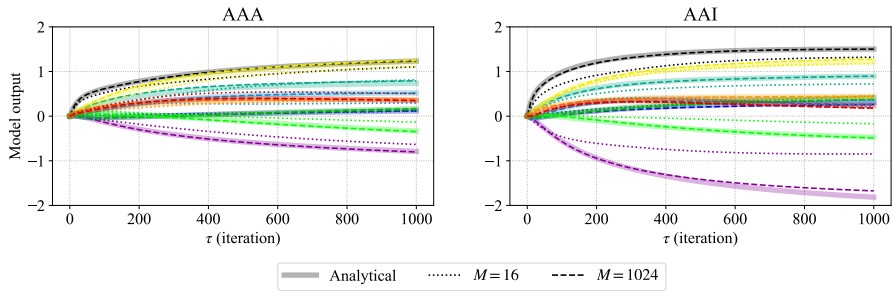

Figure 4: Output dynamics of test data points for axis-aligned soft tree ensembles with two conditions. (Left): AAA, (Right): AAI. Each data point is represented by a different line color. The left and right figures are created using exactly the same training and test data.

Since we are considering axis-aligned constraints, only a single feature is used at each split for every input path $a_\ell$. It is straightforward to extend this formulation and allow multiple features at each split. In an extreme case, if all features are always used at every split, this formula matches the formulation for arbitrary soft trees without axis-aligned constraints in Theorem 1.

The difference between AAA and AAI is whether partial features of inputs are used or all features are used in $\Sigma_{\{i,j\},s}$. In AAA, the impact of features that are not used for splitting is completely ignored, while in AAI, the kernel is affected by all features through the inner product of the inputs.

Figure 3 shows $\Theta^{\text{AxisAligned}}(\boldsymbol{x}_i, \boldsymbol{x}_j)$. We set $\alpha = 2.0$ and $\beta = 0.5$. We calculated the kernel values for two rotated vectors: $\boldsymbol{x}_i = (\cos(\omega), \sin(\omega))$, $\boldsymbol{x}_j = (\cos(\omega + \phi), \sin(\omega + \phi))$ where $\omega \in [0, \pi/2]$ and $\phi \in [0, \pi]$. The line colors show $\omega$, and the x-axis shows $\phi$. We use an oblivious tree with depth 2, where we use the first feature at both depths 1 and 2 for the architecture (A) (left column), and we use the first feature at depth 1 and the second feature at depth 2 for (B) (right column). We can see that rotational invariance for the input has disappeared. This is different from the NTK induced by typical soft tree ensembles, shown by the dotted lines (Theorem 1). Moreover, when we compare the left and right plots, we can see that the kernel varies depending on the features used for splitting. In the Appendix D.1 and D.2, we show how the NTK induced by a finite number of trees converges to the limiting NTK as the number of trees increases, and the visualization of the kernel when changing hyperparameters.

Figure 4 shows that for both AAA and AAI, as the number of trees increases, the trajectory obtained analytically from the limiting kernel and the trajectory during gradient descent training become more similar. This result validates the use of the NTK framework for analyzing training behavior. For our experiment, we consider an ensemble of oblivious trees with $\alpha = 2.0, \beta = 0.5$, where the first feature is used for splitting at depth 1 and the second feature at depth 2. The training and test datasets contain 10 randomly generated $F = 2$ dimensional points each. The prediction targets are also randomly generated. The models with $M = 16$ and $1024$ are trained using full-batch gradient descent with a learning rate of $0.1$. The initial outputs are shifted to zero (Chizat et al., 2019). Based on Lee et al. (2019), to derive analytical trajectories, we use the limiting kernel (Theorem 2), as $f(\boldsymbol{\nu}, \boldsymbol{\theta}_\tau) = \boldsymbol{H}(\boldsymbol{\nu}, \boldsymbol{x})\boldsymbol{H}(\boldsymbol{x}, \boldsymbol{x})^{-1}(\boldsymbol{I} - \exp[-\eta\boldsymbol{H}(\boldsymbol{x}, \boldsymbol{x})\tau])\boldsymbol{y}$, where $\boldsymbol{H}$ is a function that returns the limiting NTK matrix for two input matrices, and $\boldsymbol{I}$ represent an identity matrix. The input vector $\boldsymbol{\nu} \in \mathbb{R}^{F \times 1}$ is arbitrary, and the training dataset and targets are denoted by $\boldsymbol{x} \in \mathbb{R}^{F \times N}$ and $\boldsymbol{y} \in \mathbb{R}^N$, respectively. The behavior of the prediction trajectory changes depending on the configurations (AAA or AAI), even when exactly the same training and test data are used. In Appendix D.3, we present results from a real-world dataset, where one can see a similar trend.

### 3.3 THE NTK INDUCED BY ENSEMBLES OF VARIOUS TREES

In Section 3.2, we have followed the prior work (as per Theorem 1) and assumed that soft tree ensembles consist of weak learners with identical architectures, as shown on the left-hand side of Figure 5. However, it is more practical if tree structures and features for splitting vary within an ensemble, as illustrated on the right-hand side of Figure 5. To address this issue, we theoretically analyze ensembles with various tree architectures mixed together. Assuming the existence of an

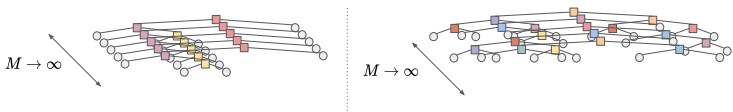

Figure 5: Ensemble of trees with different tree architectures. The color of tree nodes indicates a feature used for splitting.

infinite number of trees for each architecture in an ensemble, the NTK can be computed analytically using the amount (ratio) of each architecture in the ensemble.

**Proposition 1.** *For any input $\boldsymbol{x}_i$, let $p(\boldsymbol{x}_i, \boldsymbol{\theta}_\tau)$ be the sum of two model functions $q(\boldsymbol{x}_i, \boldsymbol{\theta}'_\tau)$ and $r(\boldsymbol{x}_i, \boldsymbol{\theta}''_\tau)$, where $\boldsymbol{\theta}'_\tau \in \mathbb{R}^{P'}$ and $\boldsymbol{\theta}''_\tau \in \mathbb{R}^{P''}$ are trainable parameters and $\boldsymbol{\theta}_\tau$ is the concatenation of $\boldsymbol{\theta}'_\tau$ and $\boldsymbol{\theta}''_\tau$ used as trainable parameters of p. For any input pair $\boldsymbol{x}_i$ and $\boldsymbol{x}_j$, the NTK induced by p is equal to the sum of the NTKs of q and r: $\widehat{\Theta}_\tau^{(p)}(\boldsymbol{x}_i, \boldsymbol{x}_j) = \widehat{\Theta}_\tau^{(q)}(\boldsymbol{x}_i, \boldsymbol{x}_j) + \widehat{\Theta}_\tau^{(r)}(\boldsymbol{x}_i, \boldsymbol{x}_j)$.*

For example, let $q$ and $r$ be functions that represent perfect binary tree ensemble models with a depth of 1 and 2, respectively. In this case, the NTK induced by trees with a depth of 1 and 2 is the sum of the NTK induced by trees with a depth of 1 and the NTK induced by trees with a depth of 2. Note that one can straightforwardly generalize it to ensembles containing various tree architectures.

Proposition 1 is particularly relevant in the context of axis-aligned trees, as it is impractical to have identical features for splitting across all trees. In addition, this proposition is applicable to not only tree ensembles but also various other models. For example, the Neural Additive Model (NAM) (Agarwal et al., 2021), which is a GAM using neural networks, can be treated using this proposition.

# 4 INSIGHTS DERIVED FROM THE NTK INDUCED BY AXIS-ALIGNED TREES

In this section, we present insights obtained using the NTK induced by the axis-aligned tree ensembles.

## 4.1 SUFFICIENCY OF THE OBLIVIOUS TREE FOR ARCHITECTURE CANDIDATES

The oblivious tree architecture is a practical design where the decision rules for tree splitting are shared across the same depth. This approach reduces the number of required splitting calculations from an exponential time and space complexity of $\mathcal{O}(2^D)$ to a linear time complexity of $\mathcal{O}(D)$, where $D$ represents the depth of the perfect binary tree. This property makes the oblivious tree architecture a popular choice in open-source libraries such as CatBoost (Prokhorenkova et al., 2018) and NODE (Popov et al., 2020). Kanoh & Sugiyama (2022) demonstrated that parameter sharing used in oblivious trees does not affect the NTK of soft tree ensembles. However, their analysis does not give any insight if only specific features are used for each splitting node.

With Theorem 2 and Proposition 1, we show that we can always convert axis-aligned non-oblivious tree ensembles into axis-aligned oblivious tree ensembles that induce exactly the same limiting NTK.

**Proposition 2.** *For any ensemble of infinitely many axis-aligned trees with the same architecture, one can always construct an ensemble of axis-aligned oblivious trees that induces the same limiting NTK, up to constant multiples.*

This proposition means that there is no need to consider combinations of complex trees, and it is sufficient to consider only combinations of oblivious trees. This insight supports the validity of using oblivious trees when using axis-aligned soft trees, as in Chang et al. (2022). Although various trial-and-error processes are necessary for model selection to determine features used at each node, this finding can reduce the number of processes by excluding non-oblivious trees from the search space.

Technically, the conversion to oblivious trees is achieved by creating $2^{D-1}$ copies of the tree as shown in Figure 6. Detailed explanations are provided in the Appendix C. Note that creating copies multiplies the NTK values by a constant factor, but theoretically, this does not have a significant

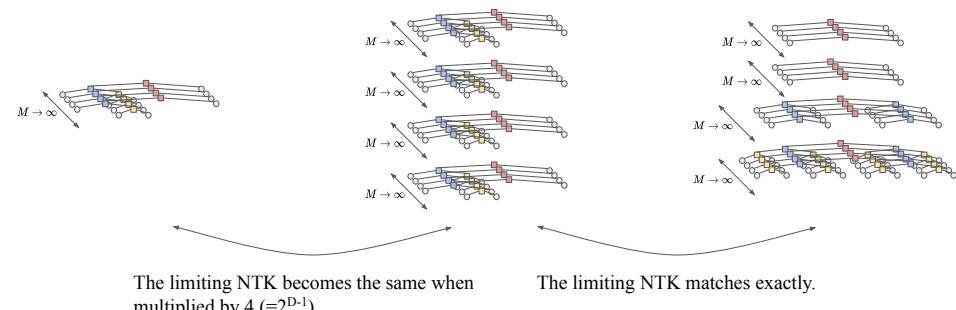

The limiting NTK becomes the same when multiplied by 4 ($=2^{D-1}$).

The limiting NTK matches exactly.

Figure 6: Conversion to oblivious trees inducing exactly the same NTK. The color of tree nodes indicates a feature used for splitting.

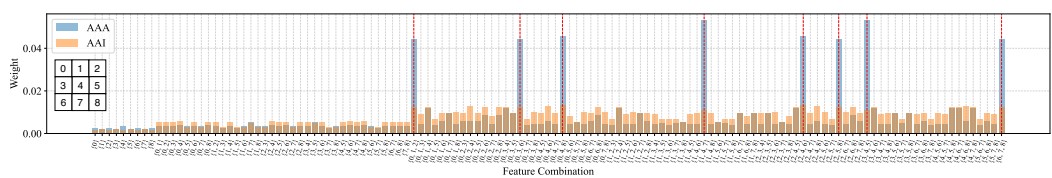

Figure 7: Weights of a linear combination of multiple kernels obtained using EasyMKL. The interactions highlighted by red dotted vertical lines indicate the feature combinations that determine the outcome of the tic-tac-toe game. The correspondence between the game board and the feature indices is displayed on the left side of the figure.

impact. As Equation 5 shows, even if two kernels are different only up to a constant, adjusting the learning rate $\eta$ can make their training behavior exactly the same.

## 4.2 MULTIPLE KERNEL LEARNING AS TREE ARCHITECTURE SEARCH

Our theoretical analysis in Section 3 assumes that features used at nodes are predetermined. To alleviate this limitation and include feature selection, we use MKL (Gönen & Alpaydın, 2011). MKL determines the weights $\rho_i$ of a linear combination of multiple kernels via training such that $\sum_i \rho_i = 1$ and $\rho_i \geq 0$ for all $i$. Using NTKs induced by various tree architectures in MKL, it becomes possible to learn how much each tree architecture should exist (Proposition 1). This approach can be also interpreted as Neural Architecture Search (NAS) (Elsken et al., 2019; Chen et al., 2021a; Xu et al., 2021; Mok et al., 2022).

We use EasyMKL (Aiolli & Donini, 2015), a convex approach that identifies kernel combinations maximizing the margin between classes. Figure 7 displays the weights obtained by EasyMKL on the entire tic-tac-toe dataset[1] preprocessed by Fernández-Delgado et al. (2014). Tic-tac-toe is a two-player game in which the objective is to form a line of three consecutive symbols (either "X" or "O") horizontally, vertically, or diagonally on a $3 \times 3$ grid. The tic-tac-toe dataset provides the status for each of the $F = 3 \times 3 = 9$ positions, indicating whether an "X" or "O" is placed or if it is blank, and classifies the outcome based on this data. We enumerate all the combination patterns from the first to the third order and use EasyMKL to determine the linear combination of $\binom{F}{1} + \binom{F}{2} + \binom{F}{3} = 129$ kernels[2] with $\alpha = 2.0$ and $\beta = 0.5$. As shown in Figure 7, for AAA, interactions that are essential to determine the outcome of the game carry significant weights. In contrast, for AAI, weights tend to be more uniform. This suggests that AAA is more sensitive to the nature of data than AAI, while a simple approach that randomly selects diverse tree architectures can be effective for AAI in practice.

---

[1] https://archive.ics.uci.edu/dataset/101/tic+tac+toe+endgame

[2] While there are more than 129 tree architecture candidates when considering factors such as which feature is used for the initial splitting, as shown in Proposition 2, we only need to consider the oblivious tree. Therefore, the order of splitting becomes irrelevant to the model's behavior. As a result, considering the 129 patterns ensures that we cover all the possible patterns.

Such a trend appears to hold true on not only the tic-tac-toe dataset but across a wide range of datasets. Details can be found in the Appendix D.4.

We also analyze the generalization performance on the tic-tac-toe dataset. Three types of the limiting NTKs induced by the soft tree ensembles are employed: AAA, AAI (Theorem 2) and Oblique (Theorem 1), as shown in Figure 2. For the oblique kernel, we assumed a perfect binary tree structure and, since AAA and AAI consider interactions up to the third order, we set the tree depth to 3. The Support Vector Machine (SVM)[3] with these kernels was used for classification (Hearst et al., 1998). Kernel parameters were set with $\alpha$ in $\{0.5, 1.0, 2.0, 4.0\}$ and $\beta$ in $\{0.1, 0.5, 1.0\}$. We used the regularization strength $C = 1.0$ in SVMs. It is known that an SVM using the NTK is equivalent to the training of its corresponding model (Chen et al., 2021b). For both AAA and AAI, a total of $\binom{F}{1} + \binom{F}{2} + \binom{F}{3} = 129$ kernels were prepared and three types of weights for the linear combination of these kernels were tested: the weight of the first type, called "MKL", is obtained by EasyMKL; the second, called "Optimal", is $1/8$ if the interaction determines the outcome of the tic-tac-toe game (there are eight such interactions) and $0$ otherwise; and the third, called "Benchmark", is uniform for all kernels. Additionally, we present the performance of Random Forest[4] (Breiman, 2001) with `max_depth = 3` and `n_estimators = 1000`.

Figure 8 displays the results of four-fold cross-validation, where 25% of the total amount of data was used for training and the remainder for evaluation. No significant variations were observed when adjusting $\beta$, so we present results with $\beta = 0.5$. Detailed experimental results, including those obtained by modifying $\beta$ and comparisons with forest models under diverse configurations, are provided in the Appendix D.5 and D.6. From the results, it is evident that setting appropriate weights for each interaction using methods like MKL improves generalization performance. This improvement is particularly remarkable in AAA, that is, AAA (MKL) and AAA (Optimal) are superior to AAA (Benchmark) across all $\alpha$. For AAI, the performance is comparable between AAI (Optimal) and AAI (Benchmark), which is consistent with the insights obtained from Figure 7. Under the

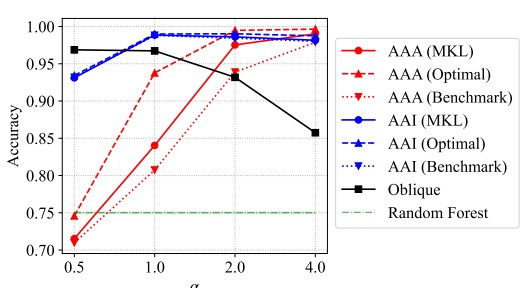

Figure 8: Classification accuracy on the tic-tac-toe dataset. The performance of Random Forest does not depend on $\alpha$, and is represented by the horizontal line. Since only Random Forest has randomness, its standard deviation of the four-fold cross-validation accuracy in 12 executions is shown by a semi-transparent band.

optimal hyperparameters, the performance was ranked in the order of AAA, AAI, and then Oblique. This order reflects the strength of the inductive bias. While AAA may be more challenging than AAI in terms of feature selection, the effort might be worthwhile if one is striving for maximum performance. Moreover, since $\alpha$ adjusts the proximity of the sigmoidal decision function to the step function, our result of different optimal values of $\alpha$ between AAA, AAI, and Oblique means that the optimal structure of the decision function can vary depending on training constraints, which is a noteworthy insight.

## 5 CONCLUSION

In this paper, we have formulated the NTK induced by the axis-aligned soft tree ensembles, and we have succeeded in describing the analytical training trajectory. We have theoretically analyzed two scenarios, one where the axis-aligned constraint is applied throughout the training process, and the other where the initial model is axis-aligned and training proceeds without any constraints. We have also presented a theoretical framework to deal with non-identical tree architectures simultaneously and used it to provide theoretical support for the validity of using oblivious trees. Furthermore, through experiments using MKL, we have shown that the suitable features for AAA and AAI can be different from each other. Our work highlights the importance of understanding the impact of tree architecture on model performance and provides insights into the design of tree-based models.

---

[3] https://scikit-learn.org/stable/modules/generated/sklearn.svm.SVC.html

[4] https://scikit-learn.org/stable/modules/generated/sklearn.ensemble.RandomForestClassifier.html

ETHICS STATEMENT

Our perspective is that the theoretical examination of the NTK will not result in detrimental uses.

REPRODUCIBILITY STATEMENT

All proofs can be found in the Appendix. To reproduce the numerical tests and illustrations, the source codes are available in the supplementary material.

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

## A PROOF OF THEOREM 2

**Theorem 2.** *Assume that all $M$ trees have the same tree architecture. Let $\{a_1, \cdots, a_\ell, \cdots, a_\mathcal{L}\}$ denote the set of decomposed paths of the trees from the root to the leaves, and let $h(a_\ell) \subset \mathbb{N}$ be the set of feature indices used in splits of the input path $a_\ell$. For any tree architecture, as the number of trees $M$ goes to infinity, the NTK probabilistically converges to the following deterministic limiting kernel:*

$$\Theta^{\text{AxisAligned}}(\boldsymbol{x}_i, \boldsymbol{x}_j) \coloneqq \lim_{M \to \infty} \widehat{\Theta}_0^{\text{AxisAligned}}(\boldsymbol{x}_i, \boldsymbol{x}_j)$$

$$= \sum_{\ell=1}^{\mathcal{L}} \left( \sum_{s \in h(a_\ell)} \Sigma_{\{i,j\},s} \dot{\mathcal{T}}_{\{i,j\},s} \prod_{t \in h(a_\ell) \setminus \{s\}} \mathcal{T}_{\{i,j\},t} + \prod_{s \in h(a_\ell)} \mathcal{T}_{\{i,j\},s} \right), \quad \text{(A.1)}$$

where $\mathcal{T}_{\{i,j\},s} = \mathbb{E}[\sigma(ux_{i,s} + \beta v)\sigma(ux_{j,s} + \beta v)]$ and $\dot{\mathcal{T}}_{\{i,j\},s} = \mathbb{E}[\dot{\sigma}(ux_{i,s} + \beta v)\dot{\sigma}(ux_{j,s} + \beta v)]$. Here, scalars $u, v \in \mathbb{R}$ are sampled from zero-mean i.i.d. Gaussians with unit variance. For $\Sigma_{\{i,j\},s}$, it is $x_{i,s}x_{j,s} + \beta^2$ when AAA is used, and $\boldsymbol{x}_i^\top \boldsymbol{x}_j + \beta^2$ when AAI is used. Furthermore, if the decision function is the scaled error function, $\mathcal{T}_{\{i,j\},s}$ and $\dot{\mathcal{T}}_{\{i,j\},s}$ are obtained in closed-form as

$$\mathcal{T}_{\{i,j\},s} = \frac{1}{2\pi} \arcsin \left( \frac{\alpha^2(x_{i,s}x_{j,s} + \beta^2)}{\sqrt{(\alpha^2(x_{i,s}^2 + \beta^2) + 0.5)(\alpha^2(x_{j,s}^2 + \beta^2) + 0.5)}} \right) + \frac{1}{4}, \quad \text{(A.2)}$$

$$\dot{\mathcal{T}}_{\{i,j\},s} = \frac{\alpha^2}{\pi} \frac{1}{\sqrt{\left(1 + 2\alpha^2(x_{i,s}^2 + \beta^2)\right)\left(1 + 2\alpha^2(x_{j,s}^2 + \beta^2)\right) - 4\alpha^4(x_{i,s}x_{j,s} + \beta^2)^2}}. \quad \text{(A.3)}$$

*Proof.* Based on the independence of parameters at each leaf and the symmetry of the decision function, Kanoh & Sugiyama (2023) showed that the NTK induced by arbitrary soft tree ensembles can be decomposed into the sum of the NTKs induced by the rule sets, which are constructed by paths from the tree root to leaves. This property of the independence of parameters at each leaf and the symmetry of the decision function also holds in our formulation (Section 2.1). Therefore, we formulate the NTK induced by rule sets and use it to derive the NTK induced by axis-aligned soft tree ensembles.

For simplicity, first we assume $\beta = 0$ in Equation 1. Let $D_\ell$ be the depth of a rule set, which is a path from the root to a leaf $\ell$. We consider the contribution from internal nodes $\Theta^{(D_\ell, \text{Rule,nodes})}$ and the contribution from leaves $\Theta^{(D_\ell, \text{Rule,leaves})}$ separately, such that

$$\Theta^{(D_\ell, \text{Rule})}(\boldsymbol{x}_i, \boldsymbol{x}_j) = \Theta^{(D_\ell, \text{Rule,nodes})}(\boldsymbol{x}_i, \boldsymbol{x}_j) + \Theta^{(D_\ell, \text{Rule,leaves})}(\boldsymbol{x}_i, \boldsymbol{x}_j). \quad \text{(A.4)}$$

As for internal nodes, when we treat the axis-aligned case (Section 3.1), only a single parameter in $\boldsymbol{w}_{m,n}$ is non-zero at initialization. When calculating the NTK as shown in Equation 5, the parameter derivatives in terms of trainable parameters are considered. In the cases of AAA and AAI, they are given as follows:

$$\frac{\partial f^{(D_\ell, \text{Rule})}(\boldsymbol{x}_i, \boldsymbol{w}, \boldsymbol{\pi})}{\partial w_{m,n,k_n}} = \frac{1}{\sqrt{M}} x_{i,k_n} \dot{\sigma}(w_{m,n,k_n} x_{i,k_n}) f_m^{(D_\ell - 1, \text{Rule})}(\boldsymbol{x}_i, \boldsymbol{w}_{m,-n}, \boldsymbol{\pi}_m), \quad \text{(AAA)}$$
$$\text{(A.5)}$$

$$\frac{\partial f^{(D_\ell, \text{Rule})}(\boldsymbol{x}_i, \boldsymbol{w}, \boldsymbol{\pi})}{\partial \boldsymbol{w}_{m,n}} = \frac{1}{\sqrt{M}} \boldsymbol{x}_i \dot{\sigma}(w_{m,n,k_n} x_{i,k_n}) f_m^{(D_\ell - 1, \text{Rule})}(\boldsymbol{x}_i, \boldsymbol{w}_{m,-n}, \boldsymbol{\pi}_m), \quad \text{(AAI)}$$
$$\text{(A.6)}$$

where $x_{i,k_n}$ and $w_{m,n,k_n}$ are $k_n$-th feature in $\boldsymbol{x}_i$ and $k_n$-th parameter in $\boldsymbol{w}_{m,n}$, respectively, and $\boldsymbol{w}_{m,-n}$ denotes the internal node parameter matrix except for the parameters of the node $n$.

As a preliminary step for calculating the NTK, we obtain the following equation:

$$\mathbb{E}_m \left[ f_m^{(D_\ell, \text{Rule})}(\boldsymbol{x}_i, \boldsymbol{w}_m, \boldsymbol{\pi}_m) f_m^{(D_\ell, \text{Rule})}(\boldsymbol{x}_j, \boldsymbol{w}_m, \boldsymbol{\pi}_m) \right]$$

$$= \mathbb{E}_m \left[ \sigma(\boldsymbol{w}_{m,1}^\top \boldsymbol{x}_i)\sigma(\boldsymbol{w}_{m,1}^\top \boldsymbol{x}_j) \cdots \sigma(\boldsymbol{w}_{m,D}^\top \boldsymbol{x}_i)\sigma(\boldsymbol{w}_{m,D}^\top \boldsymbol{x}_j)\pi_{m,\ell}^2 \right]$$

$$= \mathbb{E}_m \left[ \underbrace{\sigma(w_{m,1,k_1} x_{i,k_1})\sigma(w_{m,1,k_1} x_{j,k_1})}_{\rightarrow \mathcal{T}_{\{i,j\},k_1}} \cdots \underbrace{\sigma(w_{m,D_\ell,k_{D_\ell}} x_{i,k_{D_\ell}})\sigma(w_{m,D_\ell,k_{D_\ell}} x_{j,k_{D_\ell}})}_{\rightarrow \mathcal{T}_{\{i,j\},k_{D_\ell}}} \underbrace{\pi_{m,\ell}^2}_{\rightarrow 1} \right]$$

$$= \prod_{t \in h(a_\ell)} \mathcal{T}_{\{i,j\},t}, \quad \text{(A.7)}$$

where the symbol "$\rightarrow$" denotes the expected value of the corresponding term will take. The transition from the second line to the third line in Equation A.7 uses the equality $\sigma(\boldsymbol{w}_{m,n}^\top \boldsymbol{x}_i) = \sigma(w_{m,n,k_n} x_{i,k_n})$.

Using Equation A.7, the limiting NTK contribution from the $n$-th node is

$$\lim_{M\to\infty} \frac{1}{M} \sum_{m=1}^{M} \Bigg( \Sigma_{\{i,j\},k_n} \times \dot{\sigma}(w_{m,n,k_n} x_{i,k_n}) \dot{\sigma}(w_{m,n,k_n} x_{j,k_n})$$

$$\times f_m^{(D_\ell-1,\text{Rule})} (\boldsymbol{x}_i, \boldsymbol{w}_m, \boldsymbol{\pi}_m) f_m^{(D_\ell-1,\text{Rule})} (\boldsymbol{x}_j, \boldsymbol{w}_m, \boldsymbol{\pi}_m) \Bigg)$$

$$= \Sigma_{\{i,j\},k_n} \times \mathbb{E}_m \Bigg[ \underbrace{\dot{\sigma}(w_{m,n,k_n} x_{i,k_n}) \dot{\sigma}(w_{m,n,k_n} x_{j,k_n})}_{\to \dot{\mathcal{T}}_{\{i,j\},k_n}} \Bigg]$$

$$\times \mathbb{E}_m \Bigg[ \underbrace{f_m^{(D_\ell-1,\text{Rule})} (\boldsymbol{x}_i, \boldsymbol{w}_m, \boldsymbol{\pi}_m) f_m^{(D_\ell-1,\text{Rule})} (\boldsymbol{x}_j, \boldsymbol{w}_m, \boldsymbol{\pi}_m)}_{\to \prod_{t\in h(a_\ell)\setminus\{k_n\}} \mathcal{T}_{\{i,j\},t}} \Bigg], \tag{A.8}$$

where $\Sigma_{i,j,k_n} = x_{i,k_n} x_{j,k_n}$ when AAA is used, and $\Sigma_{i,j,k_n} = \boldsymbol{x}_i^\top \boldsymbol{x}_j$ when AAI is used. Since there are $D_\ell$ possible locations for $n$, we obtain

$$\Theta^{(D_\ell,\text{Rule,nodes})} (\boldsymbol{x}_i, \boldsymbol{x}_j) = \sum_{s\in h(a_\ell)} \Bigg( \Sigma_{\{i,j\},s} \dot{\mathcal{T}}_{\{i,j\},s} \prod_{t\in h(a_\ell)\setminus\{s\}} \mathcal{T}_{\{i,j\},t} \Bigg). \tag{A.9}$$

For leaves, since

$$\frac{\partial f^{(D_\ell,\text{Rule})} (\boldsymbol{x}_i, \boldsymbol{w}, \boldsymbol{\pi})}{\partial \pi_{m,\ell}} = \frac{1}{\pi_{m,\ell} \sqrt{M}} f_m^{(D_\ell,\text{Rule})} (\boldsymbol{x}_i, \boldsymbol{w}_m, \boldsymbol{\pi}_m), \tag{A.10}$$

with Equation A.7, we have

$$\Theta^{(D_\ell,\text{Rule,leaves})} (\boldsymbol{x}_i, \boldsymbol{x}_j) = \prod_{s\in h(a_\ell)} \mathcal{T}_{\{i,j\},s}. \tag{A.11}$$

Combining Equation A.7 and Equation A.11, we obtain

$$\Theta^{(D_\ell,\text{Rule,nodes})} (\boldsymbol{x}_i, \boldsymbol{x}_j) = \sum_{s\in h(a_\ell)} \Bigg( \Sigma_{\{i,j\},s} \dot{\mathcal{T}}_{\{i,j\},s} \prod_{t\in h(a_\ell)\setminus\{s\}} \mathcal{T}_{\{i,j\},t} \Bigg) + \prod_{s\in h(a_\ell)} \mathcal{T}_{\{i,j\},s}. \tag{A.12}$$

When we sum up this NTK over multiple rule sets constructed by multiple leaves, it becomes the NTK of the axis-aligned soft tree ensembles:

$$\Theta^{\text{AxisAligned}}(\boldsymbol{x}_i, \boldsymbol{x}_j) = \sum_{\ell=1}^{\mathcal{L}} \Bigg( \sum_{s\in h(a_\ell)} \Sigma_{\{i,j\},s} \dot{\mathcal{T}}_{\{i,j\},s} \prod_{t\in h(a_\ell)\setminus\{s\}} \mathcal{T}_{\{i,j\},t} + \prod_{s\in h(a_\ell)} \mathcal{T}_{\{i,j\},s} \Bigg). \tag{A.13}$$

Up until this point, we have been considering the case of $\beta = 0$. It is straightforward to take the case $\beta \neq 0$ into account because, in the case of soft tree ensemble, the bias term can be represented by using an extra feature that takes a constant value $\beta$ as input. This allows us to generally express the bias term's contribution by adding $\beta^2$ to the section where the product of the inputs is calculated. $\quad\square$

## B  PROOF OF PROPOSITION 1

**Proposition 1.** *For any input $\boldsymbol{x}_i$, let $p(\boldsymbol{x}_i, \boldsymbol{\theta}_\tau)$ be the sum of two model functions $q(\boldsymbol{x}_i, \boldsymbol{\theta}'_\tau)$ and $r(\boldsymbol{x}_i, \boldsymbol{\theta}''_\tau)$, where $\boldsymbol{\theta}'_\tau \in \mathbb{R}^{P'}$ and $\boldsymbol{\theta}''_\tau \in \mathbb{R}^{P''}$ are trainable parameters and $\boldsymbol{\theta}_\tau$ is the concatenation of $\boldsymbol{\theta}'_\tau$ and $\boldsymbol{\theta}''_\tau$ used as trainable parameters of $p$. For any input pair $\boldsymbol{x}_i$ and $\boldsymbol{x}_j$, the NTK induced by $p$ is equal to the sum of the NTKs of $q$ and $r$: $\widehat{\Theta}_\tau^{(p)}(\boldsymbol{x}_i, \boldsymbol{x}_j) = \widehat{\Theta}_\tau^{(q)}(\boldsymbol{x}_i, \boldsymbol{x}_j) + \widehat{\Theta}_\tau^{(r)}(\boldsymbol{x}_i, \boldsymbol{x}_j).$*

*Proof.* The NTK induced by this model can be decomposed into the sum of the NTKs of each tree architecture as follows:

$$
\widehat{\Theta}_\tau^{(p)}(\boldsymbol{x}_i, \boldsymbol{x}_j) = \left\langle \frac{\partial p(\boldsymbol{x}_i, \boldsymbol{\theta}_\tau)}{\partial \boldsymbol{\theta}_\tau}, \frac{\partial p(\boldsymbol{x}_j, \boldsymbol{\theta}_\tau)}{\partial \boldsymbol{\theta}_\tau} \right\rangle
$$

$$
= \left( \frac{\partial p(\boldsymbol{x}_i,\boldsymbol{\theta}_\tau)}{\partial \theta'_{\tau,1}}, \cdots, \frac{\partial p(\boldsymbol{x}_i,\boldsymbol{\theta}_\tau)}{\partial \theta'_{\tau,P'}}, \frac{\partial p(\boldsymbol{x}_i,\boldsymbol{\theta}_\tau)}{\partial \theta''_{\tau,1}}, \cdots, \frac{\partial p(\boldsymbol{x}_i,\boldsymbol{\theta}_\tau)}{\partial \theta''_{\tau,P''}} \right)
\begin{pmatrix}
\frac{\partial p(\boldsymbol{x}_j,\boldsymbol{\theta}_\tau)}{\partial \theta'_{\tau,1}} \\
\vdots \\
\frac{\partial p(\boldsymbol{x}_j,\boldsymbol{\theta}_\tau)}{\partial \theta'_{\tau,P'}} \\
\frac{\partial p(\boldsymbol{x}_j,\boldsymbol{\theta}_\tau)}{\partial \theta''_{\tau,1}} \\
\vdots \\
\frac{\partial p(\boldsymbol{x}_j,\boldsymbol{\theta}_\tau)}{\partial \theta''_{\tau,P''}}
\end{pmatrix}
$$

$$
= \left( \frac{\partial q(\boldsymbol{x}_i,\boldsymbol{\theta}'_\tau)}{\partial \theta'_{\tau,1}}, \cdots, \frac{\partial q(\boldsymbol{x}_i,\boldsymbol{\theta}'_\tau)}{\partial \theta'_{\tau,P'}}, \frac{\partial r(\boldsymbol{x}_i,\boldsymbol{\theta}''_\tau)}{\partial \theta''_{\tau,1}}, \cdots, \frac{\partial r(\boldsymbol{x}_i,\boldsymbol{\theta}''_\tau)}{\partial \theta''_{\tau,P''}} \right)
\begin{pmatrix}
\frac{\partial q(\boldsymbol{x}_j,\boldsymbol{\theta}'_\tau)}{\partial \theta'_{\tau,1}} \\
\vdots \\
\frac{\partial q(\boldsymbol{x}_j,\boldsymbol{\theta}'_\tau)}{\partial \theta'_{\tau,P'}} \\
\frac{\partial r(\boldsymbol{x}_j,\boldsymbol{\theta}''_\tau)}{\partial \theta''_{\tau,1}} \\
\vdots \\
\frac{\partial r(\boldsymbol{x}_j,\boldsymbol{\theta}''_\tau)}{\partial \theta''_{\tau,P''}}
\end{pmatrix}
$$

$$
= \left( \frac{\partial q(\boldsymbol{x}_i,\boldsymbol{\theta}'_\tau)}{\partial \theta'_{\tau,1}} \cdots \frac{\partial q(\boldsymbol{x}_i,\boldsymbol{\theta}'_\tau)}{\partial \theta'_{\tau,P'}} \right)
\begin{pmatrix}
\frac{\partial q(\boldsymbol{x}_j,\boldsymbol{\theta}'_\tau)}{\partial \theta'_{\tau,1}} \\
\vdots \\
\frac{\partial q(\boldsymbol{x}_j,\boldsymbol{\theta}'_\tau)}{\partial \theta'_{\tau,P'}}
\end{pmatrix}
+ \left( \frac{\partial r(\boldsymbol{x}_i,\boldsymbol{\theta}''_\tau)}{\partial \theta''_{\tau,1}} \cdots \frac{\partial r(\boldsymbol{x}_i,\boldsymbol{\theta}''_\tau)}{\partial \theta''_{\tau,P''}} \right)
\begin{pmatrix}
\frac{\partial r(\boldsymbol{x}_j,\boldsymbol{\theta}''_\tau)}{\partial \theta''_{\tau,1}} \\
\vdots \\
\frac{\partial r(\boldsymbol{x}_j,\boldsymbol{\theta}''_\tau)}{\partial \theta''_{\tau,P''}}
\end{pmatrix}
$$

$$
= \underbrace{\left\langle \frac{\partial q(\boldsymbol{x}_i,\boldsymbol{\theta}')}{\partial \boldsymbol{\theta}'}, \frac{\partial q(\boldsymbol{x}_j,\boldsymbol{\theta}')}{\partial \boldsymbol{\theta}'} \right\rangle}_{\widehat{\Theta}_\tau^{(q)}(\boldsymbol{x}_i,\boldsymbol{x}_j)} + \underbrace{\left\langle \frac{\partial r(\boldsymbol{x}_i,\boldsymbol{\theta}'')}{\partial \boldsymbol{\theta}''}, \frac{\partial r(\boldsymbol{x}_j,\boldsymbol{\theta}'')}{\partial \boldsymbol{\theta}''} \right\rangle}_{\widehat{\Theta}_\tau^{(r)}(\boldsymbol{x}_i,\boldsymbol{x}_j)}. \tag{B.1}
$$

Here, since $q$ is not a function of $\boldsymbol{\theta}''$ and $r$ is not a function of $\boldsymbol{\theta}'$, the property of their respective derivatives being zero is used at the transition from the second line to the third line in Equation B.1.

The NTK decomposition for any number of sub-models follows by repeatedly using this property. □

## C   PROOF OF PROPOSITION 2

**Proposition 2.** *For any ensemble of infinitely many axis-aligned trees with the same architecture, one can always construct an ensemble of axis-aligned oblivious trees that induces the same limiting NTK, up to constant multiples.*

*Proof.* We prove this proposition using Figure A.1 as an instance. It is straightforward to generalize the following discussion to any trees. As shown in Theorem 2, the limiting NTK is characterized by the root-to-leaf paths. Therefore, the limiting NTK induced by an infinite number of trees shown at ① in Figure A.1 is identical to the limiting NTK induced by an infinite number of trees shown at ② in Figure A.1. For any root-to-leaf path with the length $D_\ell$, one can always construct a single oblivious tree architecture that induces exactly the same NTK using $2^{D_\ell}$ rule sets composed of the same features with the path. Since we consider binary splitting at every node, the number of children at each node is 2. Therefore, for the maximum depth $D$ of a tree, by having $2^{D-1}$ copies of the same tree ensembles (②→③ in Figure A.1), we can convert them into oblivious trees (③→④ in Figure A.1). Here, each copy is identical in terms of its graph topological structure and the features used during the initialization of splitting nodes, but their randomly initialized parameters are independent. Note that when there are $D$ copies, the NTK also becomes $D$ times larger. This

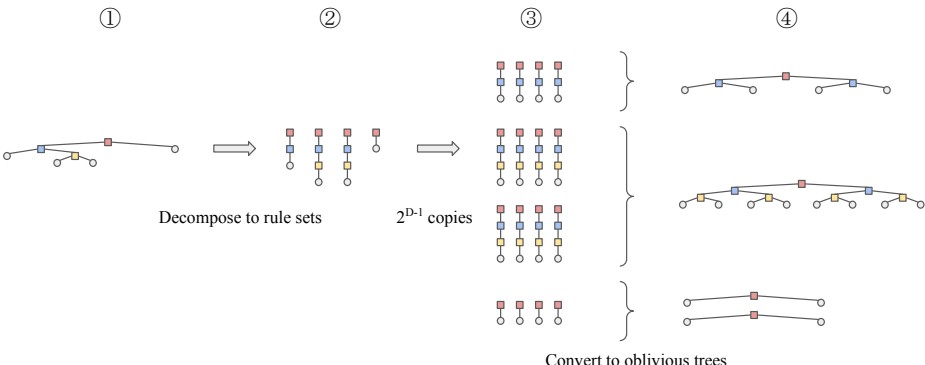

Figure A.1: A procedure to convert any arbitrary binary tree ensemble into a set of oblivious trees with the exactly same limiting NTK up to a constant multiple.

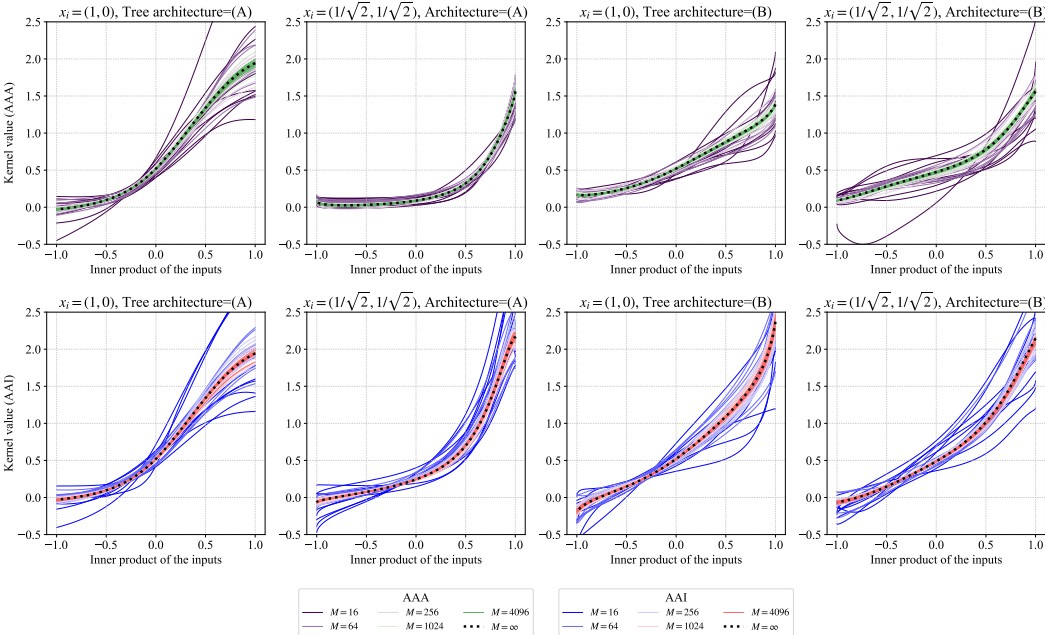

Figure A.2: An empirical demonstration of convergence of $\widehat{\Theta}_0(\boldsymbol{x}_i, \boldsymbol{x}_j)$ to the fixed limit $\Theta(\boldsymbol{x}_i, \boldsymbol{x}_j)$ as $M$ increases. Two conditions (AAA/AAI) are listed vertically, and settings of vectors and tree architectures for computing the kernel are listed horizontally.

can be understood using Equation B.1 by considering the case in Proposition 1, where we have $p(\boldsymbol{x}_i, \boldsymbol{\theta}_\tau) = q(\boldsymbol{x}_i, \boldsymbol{\theta}'_\tau) + q(\boldsymbol{x}_i, \boldsymbol{\theta}''_\tau)$. □

# D ADDITIONAL EXPERIMENTS

## D.1 CONVERGENCE OF THE NTK WITH AAA/AAI

Figure A.2 shows the convergence of the NTK with AAA or AAI cases as the number $M$ of trees increases on the same datasets and tree architectures used in Figure 3. We set $\alpha = 2.0$ and $\beta = 0.5$. The kernels induced by finite trees $M = \{16, 64, 256, 1024, 4096\}$ are computed numerically by re-initializing the parameters 10 times. We plot two cases: $\boldsymbol{x}_i = (1, 0), \boldsymbol{x}_j = (\cos(\omega), \sin(\omega))$ with $\omega \in [0, \pi]$, and $\boldsymbol{x}_i = (\frac{1}{\sqrt{2}}, \frac{1}{\sqrt{2}}), \boldsymbol{x}_j = (\cos(\omega), \sin(\omega))$ with $\omega \in [\frac{\pi}{4}, \frac{5\pi}{4}]$. We employ an oblivious tree of depth 2. For architecture (A), the first feature is used at both depths 1 and 2. For architecture

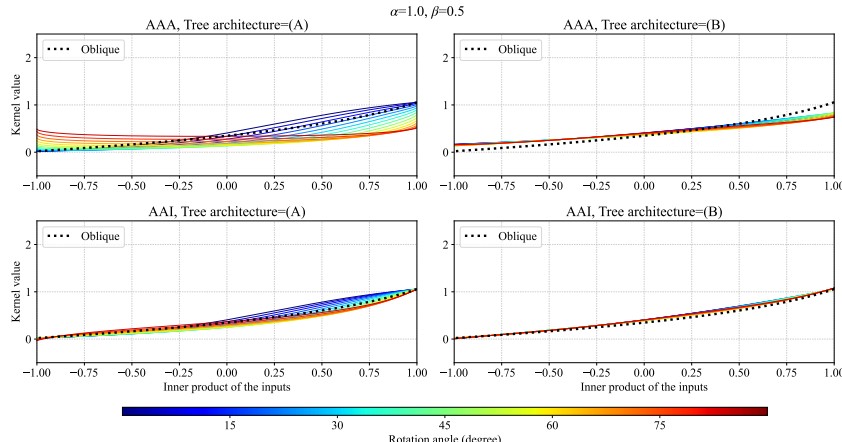

Figure A.3: The rotation angle dependency of $\Theta^{\text{AxisAligned}}(\boldsymbol{x}_i, \boldsymbol{x}_j)$ with $\alpha = 1.0$ and $\beta = 0.5$. The protocol for creating the figure is the same as Figure 3.

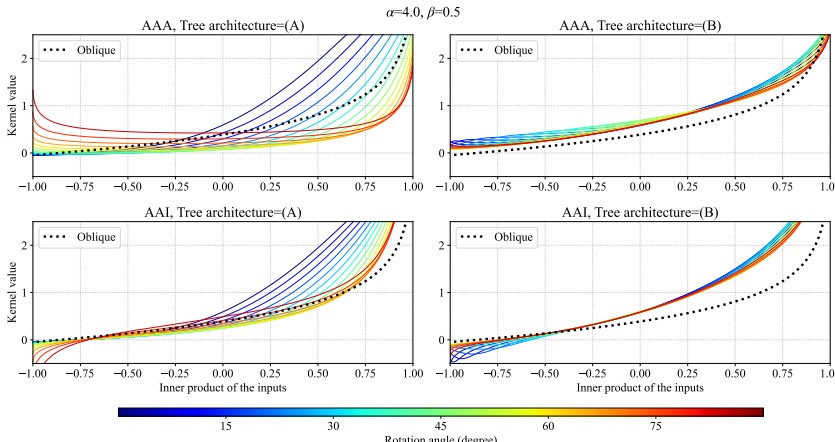

Figure A.4: The rotation angle dependency of $\Theta^{\text{AxisAligned}}(\boldsymbol{x}_i, \boldsymbol{x}_j)$ with $\alpha = 4.0$ and $\beta = 0.5$. The protocol for creating the figure is the same as Figure 3.

(B), the first feature is used at depth 1 and the second feature at depth 2. This visualization confirms that as the number of trees increases, the kernel asymptotically approaches the formula defined in Theorem 2.

### D.2 Visualization of the NTK with AAA/AAI for each hyperparameter

We performed the same visualization as in Figure 3 with varying hyperparameters. The results are shown in Figures A.3, A.4, A.5, and A.6.

### D.3 Output dynamics with a real-world dataset

Figure A.7 demonstrates that for both AAA and AAI, as the number of trees increases, the trajectory derived analytically from the limiting kernel becomes more aligned with the trajectory observed during gradient descent training. The protocol used to create this figure is the same as that for Figure 4. In this experiment, we used the diabetes dataset[5], a commonly used real-world dataset for regression tasks that predicts a quantitative measure of disease progression one year after the baseline. The diabetes dataset consists of $F = 10$ features, including body mass index, average blood pressure, age,

---

[5]https://archive.ics.uci.edu/dataset/34/diabetes

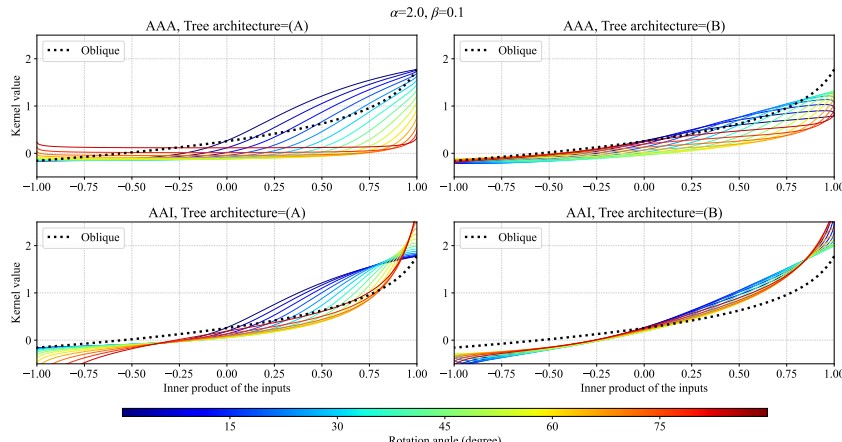

Figure A.5: The rotation angle dependency of $\Theta^{\mathrm{AxisAligned}}(\boldsymbol{x}_i, \boldsymbol{x}_j)$ with $\alpha = 2.0$ and $\beta = 0.1$. The protocol for creating the figure is the same as Figure 3.

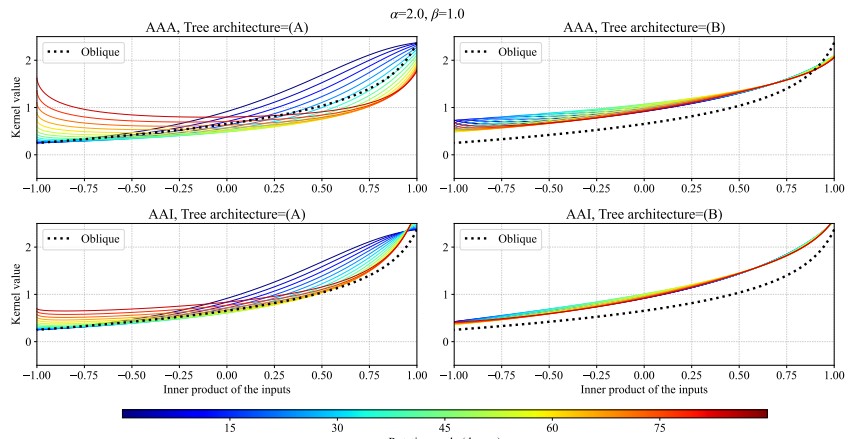

Figure A.6: The rotation angle dependency of $\Theta^{\mathrm{AxisAligned}}(\boldsymbol{x}_i, \boldsymbol{x}_j)$ with $\alpha = 2.0$ and $\beta = 1.0$. The protocol for creating the figure is the same as Figure 3.

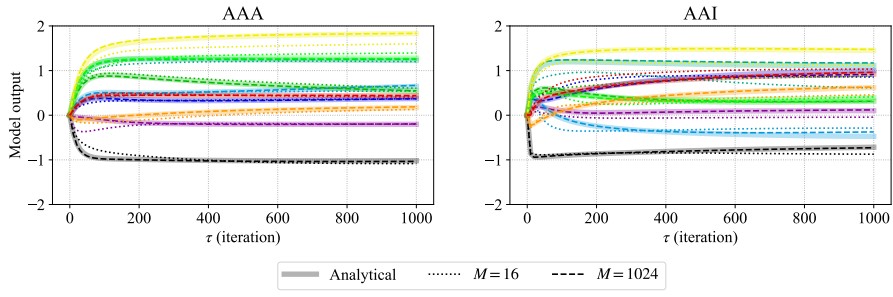

Figure A.7: Output dynamics of test data points for axis-aligned soft tree ensembles with two conditions. The protocol used to create the figure is identical to that of Figure 4
.

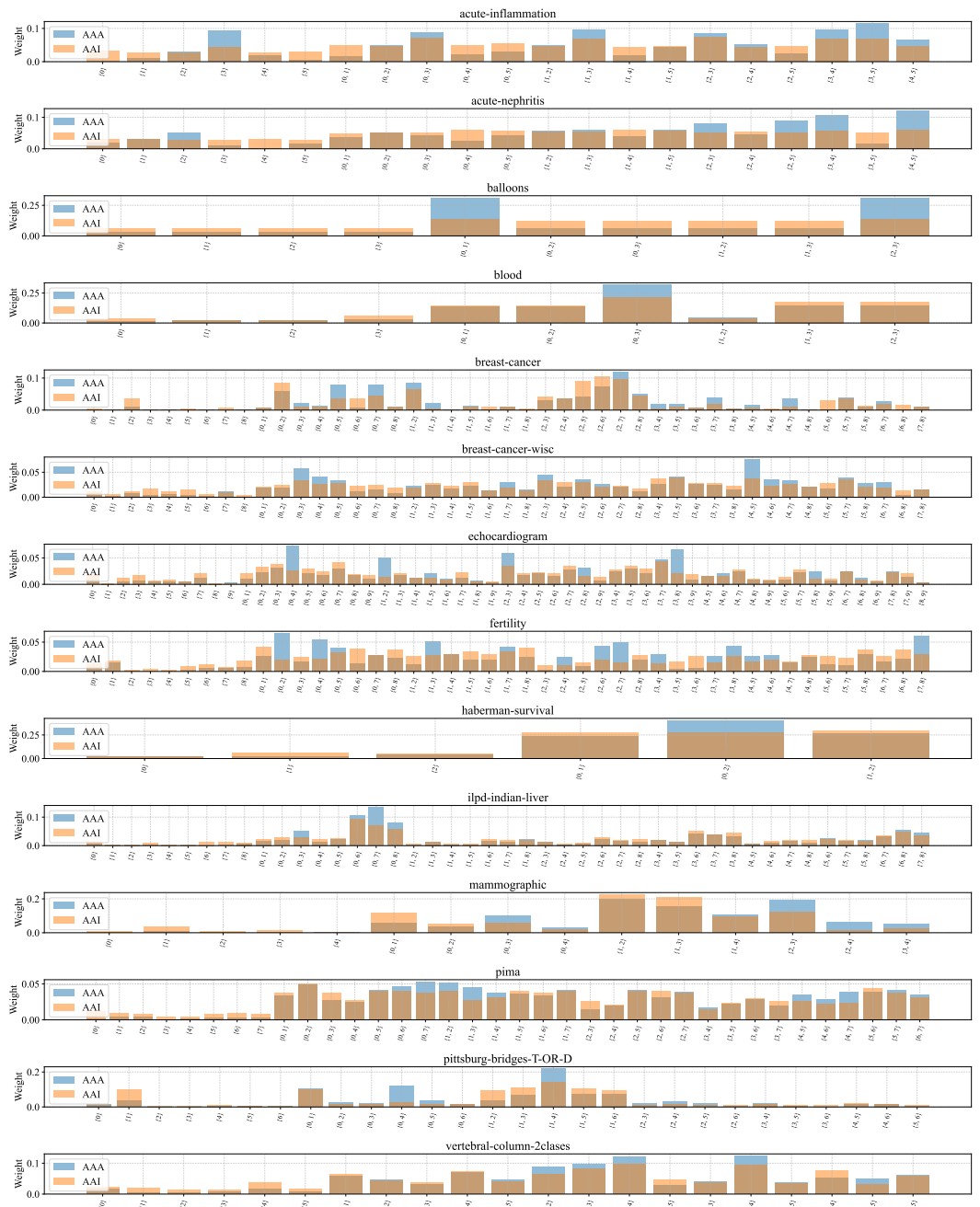

Figure A.8: Weights of a linear combination of multiple kernels obtained by EasyMKL on 14 UCI dataset.

sex, and six blood serum measurements. All features and prediction targets have been standardized to have zero mean and unit variance. We considered an ensemble of oblivious trees with parameters $\alpha = 2.0$ and $\beta = 0.5$. The body mass index was chosen for splitting at depth 1, while the average blood pressure was used for depth 2 during initialization. We selected 50 random training samples and 10 test samples for this study.

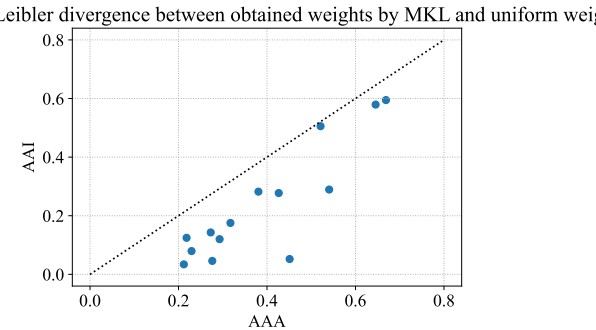

Figure A.9: The KL divergence from the weights obtained by MKL to the uniform distribution under AAA or AAI. Each point on the scatter plot corresponds to a specific dataset.

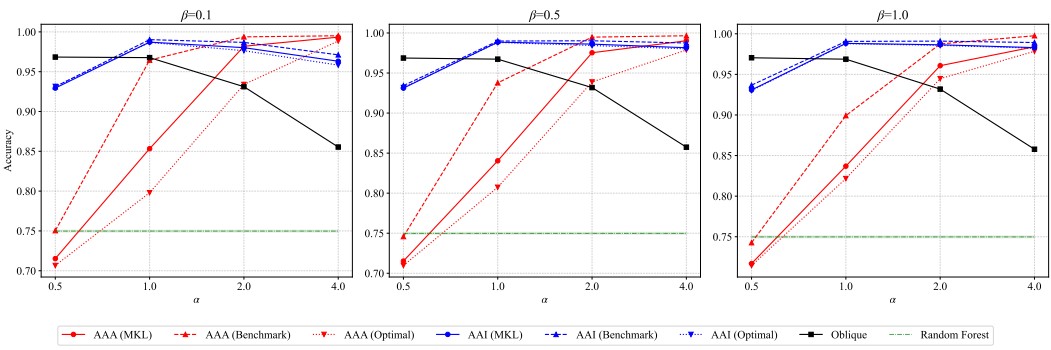

Figure A.10: Classification accuracy on the tic-tac-toe dataset with $\beta = \{0.1, 0.5, 1.0\}$. The procedure of the experiment is the same as that in Figure 8.

### D.4 MKL Weights obtained from various datasets

To investigate how MKL behaves on datasets other than the tic-tac-toe dataset, we used the UCI datasets (Dua & Graff, 2017) preprocessed by Fernández-Delgado et al. (2014). We selected and utilized 14 binary classification datasets with less than 1000 data points and 10 features. We constructed kernels considering interactions up to the second order, resulting in $\binom{F}{1} + \binom{F}{2}$ kernels. The weights obtained in a manner similar to Figure 7 are shown in Figure A.8. Similar to the tic-tac-toe dataset, AAI yields weights that are relatively close to a uniform distribution, while AAA tends to produce larger weights for specific interactions. To quantitatively analyze such trends, we compared the KL (Kullback–Leibler) divergence between the obtained weights and a uniform distribution to examine how it behaves under AAA or AAI. Results are shown in Figure A.9. From this figure, it can be seen that the KL divergence for AAA is larger, indicating a tendency of deviating from a uniform distribution, and this holds true across various datasets.

### D.5 Generalization Performance on various datasets

All the results about the generalization performances conducted in Section 4.2 are shown in Figures A.10, A.11, A.12, and A.13. As shown in Figure A.10, the accuracy trend on the tic-tac-toe dataset as shown in Figure 8 in the main text does not largely depend on $\beta$. This trend is also observed in a variety of datasets, commonly used in Appendix D.4, as shown in Figures A.11, A.12, and A.13. Overall, the performance depends on datasets and it is not fundamental to make a general claim that AAA and AAI are better or worse than other models in terms of generalization performance compared to other models. This is a natural consequence and orthogonal to our claim in this paper.

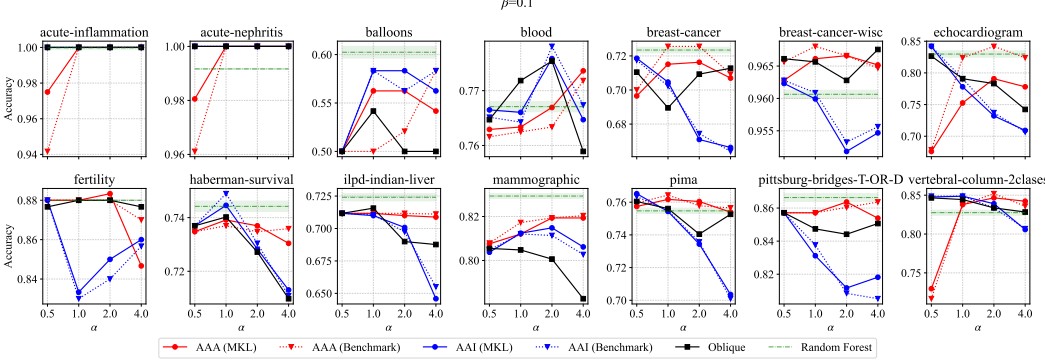

Figure A.11: Classification accuracy on 14 UCI dataset with $\beta = 0.1$. The procedure of the experiment is the same as that in Figure 8. Interactions are considered up to the second order.

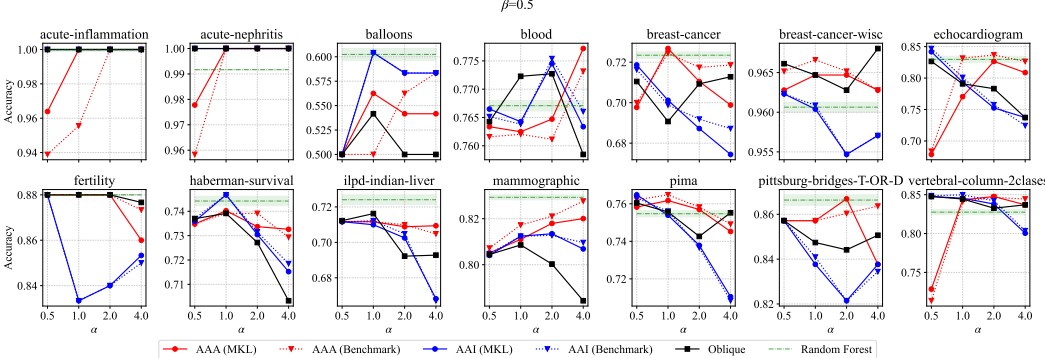

Figure A.12: Classification accuracy on 14 UCI dataset with $\beta = 0.5$. The procedure of the experiment is the same as that in Figure 8. Interactions are considered up to the second order.

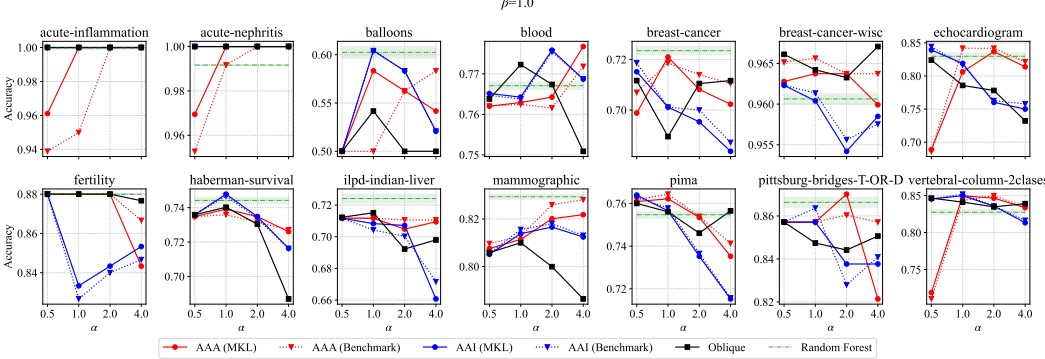

Figure A.13: Classification accuracy on 14 UCI dataset with $\beta = 1.0$. The procedure of the experiment is the same as that in Figure 8. Interactions are considered up to the second order.

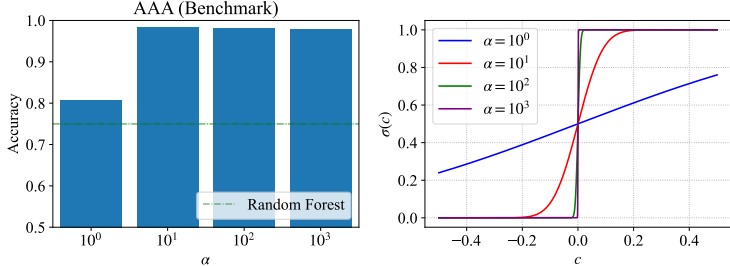

Figure A.14: Accuracy of AAA (Benchmark) on the tic-tac-toe dataset when varying $\alpha$, with $\beta = 0.5$.

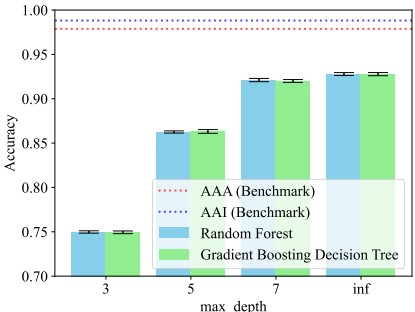

Figure A.15: Performance of Random Forest and Gradient Boosting Decision Trees on the tic-tac-toe dataset for each max_depth. The procedure of the experiment is the same as that in Figure 8.

### D.6 COMPARISON WITH RANDOM FOREST AND GRADIENT BOOSTING DECISION TREES

We further examine the performance of Random Forest and Gradient Boosting Decision Trees on the tic-tac-toe dataset and discuss the effectiveness of AAA reported in Figure 8.

Even when feature selection is not explicitly conducted ("AAA (Benchmark)" in Figure 8), the performance of the AAA model calculated using the NTK is superior to that of a typical Random Forest. In addition, as shown in Figure A.14, even if $\alpha$ increases to the extent so that the splitting function is nearly equivalent to a step function, the performance of AAA (Benchmark) remains superior to that of Random Forest. These results suggest that factors other than the feature selections and the softness of the splitting are important. This comparison of AAA (Benchmark) and Random Forest means that such a gradient descent-based learning is more effective than a greedy learning approach of Random Forest for the tic-tac-toe dataset. The output variable in the tic-tac-toe dataset, the game's outcome, is determined only by the third-order interactions of features, and it seems that greedy approaches are not appropriate to pick up such interactions.

Moreover, as shown in Figure A.15, even when the maximum depth of the tree is set to be large, the performance of Random Forest and gradient boosting[6] does not reach that of AAA (Benchmark). In terms of trained models, if the splitting function is replaced with a step function, both of AAA, Random Forest, and Gradient Boosting Decision Trees have the same format, indicating that there is no difference in the expressive power due to tree architectures. These observations indirectly support the fact that how to learn parameters in the tree is a contributing factor.

### D.7 EMPIRICAL VALIDATION OF PROPOSITION 2

Figure A.16 shows the empirical validation of Proposition 2. We consider an asymmetric tree architecture, where the first feature is used for splitting at depth 1 and the second feature used for splitting at depth 2. The left child of the first splitting node is not a splitting node but a leaf node. Using Figure A.16, we can verify whether the trajectories trained under the conditions of AAA and AAI, for $M = 16$ and $M = 1024$, respectively, match those obtained using the tree converted to

---

[6]https://scikit-learn.org/stable/modules/generated/sklearn.ensemble.GradientBoostingClassifier.html

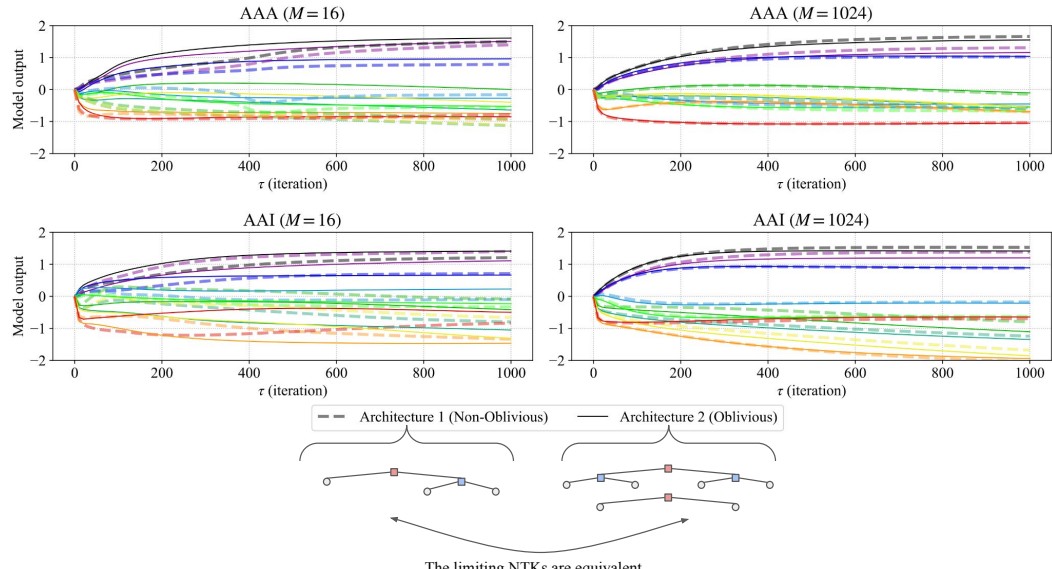

Figure A.16: Output dynamics of test data points for axis-aligned soft tree ensembles under four conditions. (Top left): AAA with $M = 16$, (Top right): AAA with $M = 1024$, (Bottom left): AAI with $M = 16$, (Bottom right): AAI with $M = 1024$. Dashed and solid lines represent the asymmetric tree model and the oblivious trees converted using Proposition 2, respectively. Each data point is represented by a different line color. All plots are created using exactly the same training and test data.

oblivious trees using Proposition 2. Results show that if there are a total of $1024$ trees, the behavior before and after the conversion is consistent. Note that in the case of architecture 2, since there are two tree architectures after conversion, each tree architecture has $8$ and $512$ trees respectively so that the total number of trees is $16$ and $1024$. The method for training a finite number of trees is the same as that in Figure 4.

# E   COMPUTATIONAL COMPLEXITY

First we analyze the computational cost of kernel matrix computation. To obtain a single kernel matrix, a calculation defined in Equation 9 is performed $N^2$ times, where $N$ is the size of an input dataset. When we denote the number of leaves as $\mathcal{L}$ and the depth of the tree as $D$, the overall computational complexity is $\mathcal{O}(N^2 \mathcal{L} D^2)$. Note that if there are duplicates in the output of $h(a_\ell)$ for all $\ell \in [\mathcal{L}]$, the computational cost can be reduced by aggregating and computing these duplicates together, so the actual computational cost is often less than this. For example, when considering oblivious trees, the computational complexity reduces to $\mathcal{O}(N^2 D^2)$.

Second, we consider the computational cost of MKL. Since MKL uses multiple kernels, it repeats calculation of kernel matrices for all the kernel matrices, while parallelization is possible in this process. Note that if there are duplicates in the output of $h(a_\ell)$ for all $\ell \in [\mathcal{L}]$, the computational cost can actually be reduced by aggregating and computing these duplicates together, so the actual computational cost is often less than this. The cost to calculate the weights of the kernels by EasyMKL depends on the number of kernels. Specifically, the EasyMKL uses Kernelized Optimization of the Margin Distribution (KOMD) (Aiolli et al., 2008), and its computational cost is known to be linear with respect to the number of kernels (Aiolli & Donini, 2015).

