# OpenReview forum: "Neural Tangent Kernels for Axis-Aligned Tree Ensembles"
_ICLR.cc/2024/Conference — Submitted to ICLR 2024_

### Official Review · Reviewer_Nxzp · 2023-10-25

**Soundness:** 3 good
**Presentation:** 4 excellent
**Contribution:** 3 good
**Rating:** 6
**Confidence:** 3

**Summary:**

This paper builds upon the theoretical analysis framework introduced by Kanoh & Sugiyama (2022; 2023) to investigate the learning behavior of soft tree ensembles. It focuses on deriving closed-form solutions for the Neural Tangent Kernel (NTK) induced by infinitely many axis-aligned tree ensembles, considering two scenarios: always axis-aligned (AAA) and axis-aligned at initialization (AAI). Additionally, the paper extends the NTK framework to accommodate multiple tree architectures and demonstrates that any non-oblivious axis-aligned tree ensemble can be transformed into an axis-aligned oblivious tree ensemble while preserving the same NTK. The paper also explores the potential applications of multiple kernel learning (MKL) in identifying suitable tree architectures and features under the axis-aligned constraint. Empirical results are provided to validate the theoretical findings and illustrate the practical significance of the axis-aligned constraint in tree ensemble learning.

**Strengths:**

1. The paper is original in extending the NTK framework to the axis-aligned soft tree ensembles, which have not been theoretically analyzed before.

2. The paper is of high quality in deriving the closed form solution of the NTK for both AAA and AAI cases, and proving the equivalence between axis-aligned non-oblivious and oblivious tree ensembles.

3. The paper is clear in presenting its main results and providing intuitive explanations and illustrations for its theoretical findings.

**Weaknesses:**

1. In Section 3.1, if I interpret it correctly, AAA refers to axis-aligned initialization, with the assumption that the selected split feature at each node remains constant throughout training. This assumption appears rather restrictive, considering that typical axis-aligned trees do not impose such constraints. It would be valuable if the author provided further insights or comments regarding this assumption.

2. The paper lacks an exploration of the computational complexity and scalability issues related to employing Multiple Kernel Learning (MKL) for tree architecture search. It is crucial to assess the feasibility and efficiency of utilizing MKL, particularly for large-scale datasets and high-dimensional feature spaces.

**Questions:**

Please find my main concerns in the weakness part. Additionally, I have another question regarding the analysis framework: does this analysis framework allow non-zero initialized parameters (selected split features) become zero (no split at one node) during the training? If the framework allows for such adaptability, it could potentially accommodate changes in the shapes or depths of trees during training, thus enhancing its overall generality.

---

> ### Author Response · Authors · 2023-11-17
> **Authors’ Response to Reviewer Nxzp**
>
> Thank you for your review.
>
> > In Section 3.1, if I interpret it correctly, AAA refers to axis-aligned initialization, with the assumption that the selected split feature at each node remains constant throughout training. This assumption appears rather restrictive, considering that typical axis-aligned trees do not impose such constraints. It would be valuable if the author provided further insights or comments regarding this assumption.
>
> Our idea is to divide the learning process of decision trees into two components:
>
> 1. selecting tree topological structure and split features,
> 2. tuning parameters for splitting thresholds at internal nodes and leaves,
>
> and providing theoretical analysis for AAA and AAI about the second component with Theorem 2. Since it can be combined with any method for the first component, the entire model is no longer restrictive in practice. Similar to the typical decision trees, it is possible to grow a tree by trying out several splitting patterns and adopting the ones with better performance. In our experiments in Section 4.2, we have used MKL to learn tree architectures for the first component and showed its effectiveness.
>
>
> > The paper lacks an exploration of the computational complexity and scalability issues related to employing Multiple Kernel Learning (MKL) for tree architecture search. It is crucial to assess the feasibility and efficiency of utilizing MKL, particularly for large-scale datasets and high-dimensional feature spaces.
>
> The overall computational cost of MKL can be divided in two parts: computation of the kernel matrix and optimization of weights.
>
> To obtain a single kernel matrix, a calculation defined in Equation (9) is performed $N^2$ times, where $N$ is the size of an input dataset. When we denote the number of leaves as $\mathcal{L}$ and the depth of the tree as $D$, the worst case overall computational complexity is $\mathcal{O}(N^2 \mathcal{L}D^2)$. Additionally, this calculation is needed to be repeated for all the kernel matrices, while parallelization is possible in this process. Note that if there are duplicates in the output of $h(a_\ell)$ for all $\ell \in [\mathcal{L}]$, the computational cost can be reduced by aggregating and computing these duplicates together, so the actual computational cost is often less than this. For example, when considering oblivious trees, the computational complexity reduces to $O(N^2 D^2)$.
>
> The cost to calculate the weights of the kernels by EasyMKL depends on the number of kernels. Specifically, EasyMKL uses Kernelized Optimization of the Margin Distribution (KOMD), and its computational cost is known to be linear with respect to the number of kernels [1].
>
> We have added the above discussion about the computational complexity in the Appendix (Section E) in our revision.
>
> Please note that MKL is mainly chosen for a better understanding of the behavior of AAI and AAI, and we have successfully shown that suitable features can differ between AAA and AAI in Figure 7. If one would like to construct more practical methods, other approaches should be considered, while it is beyond the scope of our paper.
>
> [1] Fabio Aiolli and Michele Donini (2015), EasyMKL: a scalable multiple kernel learning algorithm
>
> > Additionally, I have another question regarding the analysis framework: does this analysis framework allow non-zero initialized parameters (selected split features) become zero (no split at one node) during the training? If the framework allows for such adaptability, it could potentially accommodate changes in the shapes or depths of trees during training, thus enhancing its overall generality.
>
> Thank you for your comment. It is indeed an interesting idea. For example, incorporating L1 regularization on weights in the training process using gradient descent might be possible to obtain space weights.
> It could also be interesting to apply a temperature-scaled sparsemax [2] to the weights. We believe these ideas are interesting future directions.
>
> [2] Martins & Astudillo (2016), From Softmax to Sparsemax: A Sparse Model of Attention and Multi-Label Classification

---

### Official Review · Reviewer_QWBa · 2023-10-31

**Soundness:** 3 good
**Presentation:** 3 good
**Contribution:** 2 fair
**Rating:** 5
**Confidence:** 4

**Summary:**

The paper presents a theoretical analysis of the Neural Tangent Kernel for axis aligned decision trees. Experiments are limited to a single dataset.

**Strengths:**

This is some work towards the goal of better understanding how to optimally train axis aligned decision trees.

**Weaknesses:**

- The paper is theoretical in nature, but its impact for practical applications seems very limited to me.
- Experimental validation is limited to one dataset. This raises concerns about the practical use of this work.
- The comparison with Random Forest in unfair because RF is limited to depth 3. It is not clear how many attributes were used for splittiong at each node for RF. If the number of attributes is small, RF needs deep trees to find the relevant features.
- There is no comparison with existing modern ensembling techniques such as Gradient Boost.

**Questions:**

- What is the difference between this paper and Kanoh and Sugiyama. "Investigating Axis-Aligned Differentiable Trees through Neural Tangent Kernels". In ICML 2023 Workshop on Differentiable Almost Everything: Differentiable Relaxations, Algorithms, Operators, and Simulators?
- What are the practical implications of knowing the NTK induced by axis-aligned tree ensembles? Does this imply that we can obtain better/faster training algorithms?

**Details Of Ethics Concerns:**

The main results of the paper (Theorem 2 and Prop 1) are also present in: R Kanoh, M Sugiyama. "Investigating Axis-Aligned Differentiable Trees through Neural Tangent Kernels. ICML 2023 Workshop on Differentiable Almost Everything: Differentiable Relaxations, Algorithms, Operators, and Simulators.

---

> ### Author Response · Authors · 2023-11-17
> **Authors’ Response to Reviewer QWBa**
>
> Thank you for your review.
>
> > The paper is theoretical in nature, but its impact for practical applications seems very limited to me.
>
> We disagree with this point. For instance, Proposition 2 guarantees that the consideration of oblivious trees is sufficient for searching tree architectures, resulting in a substantial reduction in the architecture search space in practical situations. Moreover, the insight that feature selection is not important for AAI can lead to more efficient processes by reducing unnecessary searches. We believe that acquiring the NTK that enables us to comprehend training behavior, and using it to theoretically support these characteristics in a rigorous manner, is of significant importance.
>
> > Experimental validation is limited to one dataset. This raises concerns about the practical use of this work. The comparison with Random Forest in unfair because RF is limited to depth 3.
>
> > There is no comparison with existing modern ensembling techniques such as Gradient Boost.
>
> While we show the results of only one dataset in the main text, we have already conducted numerical experiments on 14 datasets about MKL weight distributions, as written in the last sentence of the second paragraph in Section 4.2. These experiments have been already described in our initial submission before revision. In addition, while it is not directly related to the main claim of the paper, empirical analyses of the generalization performance across numerous datasets have also been added to the Appendix (Section D.5). The tic-tac-toe dataset was chosen for the main content because we think it made understanding the properties of models easier. We would like to note that, in our paper, we do not focus on the practical utility of these forest models; rather, we aim to understand their theoretical properties. It is not fundamental to make a general claim that AAA and AAI are better or worse than other models in terms of generalization performance as it depends on datasets. This is a natural consequence and orthogonal to our claim in this paper. The tic-tac-toe dataset intuitively demonstrates that methods like AAA and AAI perform better than greedy methods such as Random Forest. This discussion has been added to the Appendix (Section D.5).
>
> To further address your concern, we have added comparative results of typical forest models on the tic-tac-toe dataset in the Appendix (Section D.6), including experiments where tree depth was varied or Gradient Boosting was used. Typical forest models did not surpass AAA across all the tested settings of the tree depth or learning algorithms.
>
> > It is not clear how many attributes were used for splitting at each node for RF. If the number of attributes is small, RF needs deep trees to find the relevant features.
>
> In Random Forest, each splitting node uses a single feature to make a split, which is a typical approach, and this is the same as AAA.
>
> > What are the practical implications of knowing the NTK induced by axis-aligned tree ensembles? Does this imply that we can obtain better/faster training algorithms?
>
> Yes, it does. That is an important argument of our paper. As described in Proposition 2, it is sufficient to consider only oblivious trees as tree topological structures, which significantly reduces the search space and can lead to faster training. Furthermore, our empirical results show that feature selection holds minimal importance for AAI, which also contributes to efficient training.
>
> > What is the difference between this paper and Kanoh and Sugiyama. "Investigating Axis-Aligned Differentiable Trees through Neural Tangent Kernels". In ICML 2023 Workshop on Differentiable Almost Everything: Differentiable Relaxations, Algorithms, Operators, and Simulators?
>
> There is no concern regarding this matter. We have already communicated individually with the program chairs and senior area chairs. Due to the author's instructions, we are unable to provide further details in this forum.

---

> > ### Comment · Reviewer_QWBa · 2023-11-20
> > **I maintain my rating**
> >
> > Many of my main concerns still stand:
> > - The theoretical justification is about an infinite number of trees, from which we cannot derive the conclusion that the authors derive: "Proposition 2 guarantees that the consideration of oblivious trees is sufficient for searching tree architectures" because we cannot train infinitely many trees in practice. So the authors are overclaiming.
> > - The weight experiments on 14 datasets are not experiments on accuracy. So I still think that experiments are limited to one dataset.
> > - Also, it is not clear from the experiment which ones are the oblivious trees. Looking at table A1 (which is missing two sample t-test for comparison with the best method) I see that AAA (optimal) is the best method. Are those the oblivious trees? If none of them are, what is the point of the experiment?
> > - Seems that the authors did not understand my observation about RF even though it is well known that a single attribute is used in RF for splitting at each node, but it is selected from a larger pool. So I rephrase my observation: It is not clear how many attributes were used in the pool for selecting the splitting attribute at each node for RF. If the pool is small, RF needs deep trees to find the relevant features. So I still think the RF was misrepresented in the experiment.

---

> > > ### Author Response · Authors · 2023-11-21
> > > **Reply to Reviewer QWBa**
> > >
> > > Thank you for your reply.
> > > > The theoretical justification is about an infinite number of trees, from which we cannot derive the conclusion that the authors derive: "Proposition 2 guarantees that the consideration of oblivious trees is sufficient for searching tree architectures" because we cannot train infinitely many trees in practice. So the authors are overclaiming.
> > >
> > > Yes, the theory is constructed using the NTK induced by an infinite number of trees. However, as shown in Figure 4 and Figure A.7, predicted values obtained by finite trees and those using the limiting NTK (assuming infinitely many trees) match well. We used 1024 trees and, for example, NODE [1] used 2048 trees,  suggesting that our theoretical claims including the sufficiency of oblivious trees are realistic in practical situations.
> > >
> > > [1] Popov et al. (2020), Neural Oblivious Decision Ensembles for Deep Learning on Tabular Data,
> > >
> > > > The weight experiments on 14 datasets are not experiments on accuracy. So I still think that experiments are limited to one dataset.
> > >
> > > As mentioned in our response to you during the rebuttal period, we have already added experiments on accuracy in 14 datasets in Section D.5. There is a text in our response: “In addition, while it is not directly related to the main claim of the paper, empirical analyses of the generalization performance across numerous datasets have also been added to the Appendix (Section D.5). We would appreciate it if you could check the revised paper.
> > >
> > > > Also, it is not clear from the experiment which ones are the oblivious trees. Looking at table A1 (which is missing two sample t-test for comparison with the best method) I see that AAA (optimal) is the best method. Are those the oblivious trees? If none of them are, what is the point of the experiment?
> > >
> > > All kernels in the kernel-based methods (i.e. AAA, AAI, and Oblique) are constructed assuming oblivious tree architectures as they cover all the possible patterns of features up to the third order as written in the footnote in P.8 of the paper.
> > > Please note that Table A.1 in our initial submission has been replaced with Figure A.10 in the current revised version for better presentation.
> > >
> > > > Seems that the authors did not understand my observation about RF even though it is well known that a single attribute is used in RF for splitting at each node, but it is selected from a larger pool. So I rephrase my observation: It is not clear how many attributes were used in the pool for selecting the splitting attribute at each node for RF. If the pool is small, RF needs deep trees to find the relevant features. So I still think the RF was misrepresented in the experiment.
> > >
> > > Thank you for clarifying your question. In the numerical experiments, we have used the default settings of scikit-learn, so sqrt(n_features) number of features are consistently used at splitting. To further verify the effect of the feature candidate pool, we have also conducted Random Forest considering all the features at each splitting without any sampling, and the experimental results hardly changed. In the tic-tac-toe dataset, the accuracy improved slightly from 0.750 to 0.761, so our discussion and conclusion in our paper remain unchanged.

---

> > > > ### Comment · Reviewer_QWBa · 2023-11-21
> > > > **Better, but some issues still remain.**
> > > >
> > > > Still some issues remain:
> > > > - Figure 4 and Figure A7 show that some of the trees (not clear, are they oblivious or not?) fit the infinite limit pretty well. What about the other kind? Because Proposition 2 talks about two types of infinite trees: axis aligned oblivious and non-oblivious. If you can approximate one kind and not the other with finite trees, Proposition 2 is not useful.
> > > > - Ok, Section D5 was not present in the original submission, I just found it in the revised submission. Yes, the results look pretty good but they should be put in a table instead of a number of figures so that other people can cite your work and your results precisely, without having to approximate them from the figures. The same goes for Figure A10. Tables should also have standard deviations from which p-values could be computed to check significance by anyone interested.
> > > > - Random forest is still misrepresented by displaying results with depth 3. The original random forest is supposed to have fully grown trees. I see infinite depth RF results are much better, even though not as good as the AAA and AAI.

---

> > > > > ### Author Response · Authors · 2023-11-23
> > > > > **Reply to Reviewer QWBa**
> > > > >
> > > > > Thank you for your various suggestions, which significantly enhance our paper.
> > > > >
> > > > > > Figure 4 and Figure A7 show that some of the trees (not clear, are they oblivious or not?) fit the infinite limit pretty well. What about the other kind? Because Proposition 2 talks about two types of infinite trees: axis aligned oblivious and non-oblivious. If you can approximate one kind and not the other with finite trees, Proposition 2 is not useful.
> > > > >
> > > > > In Appendix (Section D.7), we have added an experimental validation of the behavior of predicted values when non-oblivious trees are converted to oblivious ones using Proposition 2. Please check the latest PDF. As shown in Figure A.16, if there are a total of 1024 trees, the behavior of the predicted values after conversion to oblivious trees and that before the conversion match well.
> > > > >
> > > > > As for Figure 4 and A.7, we used oblivious trees. There is a sentence: “For our experiment, we consider an ensemble of oblivious trees with $\alpha=2.0, \beta=0.5$, …” in Section 3.2 in our original submission.
> > > > >
> > > > > > Ok, Section D5 was not present in the original submission, I just found it in the revised submission. Yes, the results look pretty good but they should be put in a table instead of a number of figures so that other people can cite your work and your results precisely, without having to approximate them from the figures. The same goes for Figure A10. Tables should also have standard deviations from which p-values could be computed to check significance by anyone interested.
> > > > >
> > > > > Given the limited time remaining in the rebuttal period, addressing this point now appears difficult for us. Of course, since it is just a different method of visualizing the results, it is possible to address it in the camera-ready version.
> > > > >
> > > > > Considering the aspect of readability, we believe the current format may also be good. If we try to represent all the results of every hyperparameter for all datasets and settings in tables, it would become an enormous amount of numbers in tables and, we think, reduce the readability of the paper. Since we have also provided our code, any results in our paper can be reproducible.
> > > > >
> > > > > > Random forest is still misrepresented by displaying results with depth 3. The original random forest is supposed to have fully grown trees. I see infinite depth RF results are much better, even though not as good as the AAA and AAI.
> > > > >
> > > > > We understand your point. However, our experiments aim to empirically investigate the impact of different training algorithms on the resulting performance when the tree topological structures are aligned. Thus our paper's claims and such a comparison to the infinite depth Random Forest results are orthogonal. Also, since the Appendix contains a variety of experimental results with different tree depths, all the information is included in the paper. Since our main claim in this paper is not affected by the RF results, we believe it is better to keep the visualization as it is.

---

### Official Review · Reviewer_w3rN · 2023-11-01

**Soundness:** 3 good
**Presentation:** 3 good
**Contribution:** 3 good
**Rating:** 6
**Confidence:** 2

**Summary:**

[I made a mistake in the form.] I found out that I have accidentally checked the "First Time Reviewer" question, but in fact, I'm not. It seems that I cannot undo it now, so I'm instead writing it here.

This paper proposes a way to analyze the training dynamics of axis-aligned tree ensembles using neural tangent kernels (NTK). The idea is two-fold: (1) using soft trees and assigning proper weights to derive NTK, and (2) using multiple kernel learning (MKL) for finding suitable tree structures. The paper also shows that, from any ensemble of axis-aligned trees, one can find that of axis-aligned oblivious trees with the same limiting NTK, which justifies the use of oblivious trees.

**Strengths:**

- The paper cleverly derives a way to analyze the training dynamics of axis-aligned trees using soft trees and NTK. The idea here is to deliberately assign weights in NTK so that it represents the behavior of axis-aligned trees, which is interesting.

- The online feature selection in actual tree construction is modeled using MKL, which is also interesting.

- The paper also provides a justification regarding the use of oblivious trees based on the above framework.

**Weaknesses:**

- I did not find any weaknesses.

**Questions:**

- The paper argues that one of its contributions is including finite tree ensemble scenarios. However, in Section 4, the proposition is only on infinite trees. Why is this?

---

> ### Author Response · Authors · 2023-11-17
> **Authors’ Response to Reviewer w3rN**
>
> Thank you for your review.
>
> > The paper argues that one of its contributions is including finite tree ensemble scenarios. However, in Section 4, the proposition is only on infinite trees. Why is this?
>
> Proposition 2 focuses on the property of a deterministic closed-form formula (=limiting NTK). A deterministic closed-form kernel is derived when assuming an infinitely large number of trees, and a deterministic NTK is not induced due to initialization randomness when the number of trees is finite. As the number of trees increases, the NTK asymptotically approaches a deterministic closed-form formula.

---

### Official Review · Reviewer_PQpi · 2023-11-06

**Soundness:** 4 excellent
**Presentation:** 2 fair
**Contribution:** 3 good
**Rating:** 5
**Confidence:** 3

**Summary:**

The authors derive a Neural Tangent Kernel for axis-aligned trees, and show several extensions such as non-oblivious trees and multiple different architectures. They also run a numerical experiment on the tic-tac-toe dataset and show the empirical utility of such methods.

**Strengths:**

The authors build on a few prior papers, Kanoh and Sugiyama 2022, 2023, which establish NTK on tree ensembles.  Though I did not check the proofs carefully, the theory seems sound. Prior work on NTK for tree ensembles seemed limited to all trees having the same architecture, and each tree being oblivious, and the authors extended the theory into those regimes.

**Weaknesses:**

1. I do not really understand why we would *want* NTK for tree ensembles, or more specifically axis-aligned trees.  Typically, I look to theory because it can provide insight or intuition about why some method is working well.  I did not get any of that here.  What does this theory explain about axis-aligned trees that we did not already know?

2. There are several papers relating trees/forests to kernels.  The mondrian kernel is probably the most famous, https://dl.acm.org/doi/10.5555/2969033.2969177, though Breiman wrote about it over 20 years ago prior to his random forest paper, https://citeseerx.ist.psu.edu/viewdoc/download?doi=10.1.1.24.7078&rep=rep1&type=pdf. And we https://arxiv.org/abs/1812.00029, and others, have some work on it as well, https://ieeexplore.ieee.org/document/7373647.  Those kernels are exact, whereas this kernel is an approximation.  Given that trees/forests directly induce a kernel, I do not understand why we would want to derive an approximate kernel?

3. The primary results seem to depend on all the trees having the same architecture, and also all the features being somehow already selected?  I do not understand where the architecture or selected features come from?

4.  The results also all seem to depend on soft, rather than hard, trees. Is that because the math is easier for soft trees? A few sentences about that in the discussion would be helpful context. I do not know of any tree packages that leverage soft trees, so if people actually use them in practice (I know the papers on Neural Forests, but I do not know whether they are actually used anywhere), that would be helpful context as well.  If nobody uses soft trees, that's ok, it just a limitation, and future work might be about getting similar results on hard trees, unless it is obviously (to you) not tractable.

5. The empirical results are on a single dataset: tic-tac-toe.  It is a nice illustration.  I wonder, however, why this dataset was chosen specifically? Was it cherry-picked to have good results? Or was it because all the features are binary, and that helps for some reason? Or because depth 3 trees work well (at least the AAA/AAI ones)?

6.  For me, the fact that AAA works better than RF for certain alpha's is by far the most interesting result.  What features of the distribution is the axis-aligned NTK capturing that the RF fails to acquire? I am guessing the fact that they are depth 3 trees has something to do with it, because the RF can only handle 3 feature splits per path, and more are required to achieve Bayes optimal.  How does the NTK get around this issue, what is happening?  For me, this is by far the most interesting result, and I did not understand or see any text attempting to explain it.

**Questions:**

My main question is why/when can axis-aligned NTKs outperform axis-aligned forests?  What information can they leverage that is missed by the forests? What is the inductive bias of the NTK relative to the axis-aligned forests.  While the theory, on its own, is fine, I do not find it compelling on its own.  The arbitrarily slow convergence theorem (https://link.springer.com/article/10.1007/BF00534199) implies that any given approach will outperform another with finite data.  So, from that perspective, the point of any paper describing a new approach is to provide insight into when/why it outperforms other approaches.  Curves plotting performance vs sample size, dimensionality, or various simulation parameters can all provide insight into this issue.  If the authors can provide clean compelling explanations about when/why their NTK would/does outperform RF or other kernel forest approaches, I think it would be very interesting.  Without that, however, I am just not that interested.

---

> ### Author Response · Authors · 2023-11-17
> **Authors’ Response to Reviewer PQpi [1/2]**
>
> Thank you for your review.
>
> > I do not really understand why we would want NTK for tree ensembles, or more specifically axis-aligned trees. Typically, I look to theory because it can provide insight or intuition about why some method is working well. I did not get any of that here. What does this theory explain about axis-aligned trees that we did not already know?
>
> As stated in Section 2.2, the NTK precisely describes training behavior when an axis-aligned tree ensemble is trained through gradient methods. This ability of the NTK leads to our findings such as the sufficiency of considering oblivious trees (Proposition 2) and the realization that feature selection is not important in the case of AAI (Figure 7).
>
> > Those kernels are exact, whereas this kernel is an approximation. Given that trees/forests directly induce a kernel, I do not understand why we would want to derive an approximate kernel?
>
> The typical kernel obtained from a trained forest simply measures the similarity between data points. However, the kernel we have derived is the NTK, which allows us to analytically describe the training behavior of a model without actually conducting the training. An example of this is shown in Figure 4, and the theoretical background is presented in Section 2.2. While both fall under the category of kernels, their roles are fundamentally different and our NTK is not an approximation of the kernels you have listed.
>
> NTKs have successfully provided theoretical insights for not only the tree ensemble models but also various other models such as typical MLPs, CNNs, RNNs, and Transformers, and NTKs help validate the models. Examples include the equivalence between global average pooling in CNNs and certain data augmentations [1] and the explanation of why performance does not significantly deteriorate in ResNets with Skip Connections even for deeper models [2].
>
> We would also like to clarify the issue of 'exact/approximation.' The closed-form formula for the NTK is available when considering an infinite number of trees. Moreover, even in the finite case, the kernel can be precisely derived without any approximation. Therefore our NTK is also exact in that sense. There is a randomness due to initialization and the kernel value could vary slightly with each initialization. Nevertheless, as shown in Figure A.2, this variation converges to zero as the number of trees increases, matching the deterministic kernel given in Theorem 2. We believe that such randomness also exists in the existing kernels for trained typical forest models, for example due to bagging, and this is unrelated to the question of whether it is an approximation.
>
> [1] Li et al., (2019), Enhanced Convolutional Neural Tangent Kernels
>
> [2] Huang et al., (2020), Why Do Deep Residual Networks Generalize Better than Deep Feedforward Networks? — A Neural Tangent Kernel Perspective
>
> > The primary results seem to depend on all the trees having the same architecture, and also all the features being somehow already selected? I do not understand where the architecture or selected features come from?
>
> We believe you are referring to Theorem 2. Then you are right that our primary result (Theorem 2) assumes that all the trees are equivalent in terms of the tree topological structure and selected features. However, it does not specify how to choose the features or tree topological structure, which can be independently implemented. One can select any method for choosing the features or tree structure, and our theorem applies to any selection. Even in AAA, it is possible to develop a tree by empirically examining various splitting patterns and adopting those with better performance on training data. As demonstrated in Section 4.2, it is also feasible to learn tree architectures using MKL. There are various approaches.

---

> > ### Author Response · Authors · 2023-11-17
> > **Authors’ Response to Reviewer PQpi [2/2]**
> >
> > > The results also all seem to depend on soft, rather than hard, trees. Is that because the math is easier for soft trees? A few sentences about that in the discussion would be helpful context. I do not know of any tree packages that leverage soft trees, so if people actually use them in practice.
> >
> > The soft tree, which is also called a differentiable decision tree, is not adopted solely for its theoretical manageability. In fact, soft trees are used in various places. For example, it has been implemented in well-known open-source software like PyTorch Tabular [3], and it often becomes a topic of discussion in platforms like Kaggle. In terms of an axis-aligned model, there is also active research like [4]. We believe that conducting a theoretical analysis of such models could hold significant value. We have added a sentence about PyTorch Tabular in the introduction. Moreover, the concept of the soft tree is implicitly applied in various contexts; for instance, the method of assigning data to suitable leaves is a special case of a hierarchical Mixture-of-Experts [5].
> >
> > [3] Manu Joseph (2021), PyTorch Tabular: A Framework for Deep Learning with Tabular Data
> >
> > [4] Chang et al., (2022), NODE-GAM: Neural generalized additive model for interpretable deep learning
> >
> > [5] Jodan and Jacobs (1993), Hierarchical mixtures of experts and the EM algorithm
> >
> > > The empirical results are on a single dataset: tic-tac-toe. It is a nice illustration. I wonder, however, why this dataset was chosen specifically?
> >
> > We chose the tic-tac-toe dataset for the main content because the relevant interactions are intuitive and easy to understand.
> > Note that while we show the results of only one dataset in the main text, we have conducted numerical experiments on 14 datasets about MKL weight distributions, as written in the last sentence of the second paragraph in Section 4.2. These experiments have been already described in our initial submission before revision. In addition, while it is not directly related to the main claim of the paper, empirical analyses of the generalization performance across numerous datasets have also been added to the Appendix (Section D.5).
> >
> > > For me, the fact that AAA works better than RF for certain alpha's is by far the most interesting result. What features of the distribution is the axis-aligned NTK capturing that the RF fails to acquire? I am guessing the fact that they are depth 3 trees has something to do with it, because the RF can only handle 3 feature splits per path, and more are required to achieve Bayes optimal. How does the NTK get around this issue, what is happening? For me, this is by far the most interesting result, and I did not understand or see any text attempting to explain it.
> >
> > Thank you for expressing your interest. We have added a section in Appendix D.6 that delves deeper into this perspective. We summarize the key points in the following.
> >
> > Our analysis suggests that the observed differences in performance can be attributed more to the algorithms used for learning parameters, rather than to the tree architecture or feature selection methods in the tic-tac-toe dataset. We have performed a variety of ablation studies examining factors including feature selection, the softness of splits, tree depth, and ensemble methods (including Random Forest and Gradient Boosting Decision Tree). In all these scenarios, typical forest models employing a greedy approach for parameter determination consistently failed to surpass the performance of AAA.
> >
> > The prediction obtained through the NTK is equivalent to that by the entire tree with parameter learning through gradient descent. Compared to greedy methods, the gradient method is more effective for the tic-tac-toe dataset. This is because, in the tic-tac-toe dataset, there are no effective interactions at lower-orders, and important interactions only appear at the third-order. Therefore, one can infer that a greedy approach is unsuited.
> >
> >
> > > What is the inductive bias of the NTK relative to the axis-aligned forests.
> >
> > As mentioned above, we believe it is due to the learning algorithms. The difference lies in whether parameters are updated considering the entire tree or based on a greedy approach.

---

> > > ### Comment · Reviewer_PQpi · 2023-11-22
> > > **Additional numerical experiments**
> > >
> > > Looking at Figures A.11 - A.13, it does not seem like the NTK methods outperform classical RF on typical datasets.  New methods do not need to outperform old ones on benchmark datasets to be interesting.  But for papers to be fully in integrity, imho, they need to be super clear about it.  I understand this is not the standard policy in ML/AI papers (or papers in other disciplines), but I stand by this conviction.  For me, the sentence:
> > >
> > > > Such a trend appears to hold true on not only the tic-tac-toe dataset but across a wide range of datasets. Details can be found in the Appendix D.4.
> > >
> > > is inadequate.  Highlighting a particular dataset to make a point is great, being honest about when this point holds, is greater.
> > >
> > > What I can infer currently, based on the additional numerical experiments, is that 'greedy' methods perform poorly relative to gradient-based approaches, specifically on datasets in which individual features contain relatively little information, and much more information is in the joint.  That totally makes sense.  And, that is a point that does not require (or even obviously benefit from) NTK, rather, that point can be made simply by comparing RF to PyTorch Tabular.
> > >
> > > The key question for me to motivate this entire line of work (developing NKT for axis-aligned RF) is what insight do we gain specifically from the NKT derivation.  I still do not see it.  I see why gradient-based methods work better than RF in certain cases.  And I see that you can do it, and do neural architecture search, and MKL, etc.  And this is all great.  I just don't know why.
> > >
> > > The answer I am looking for would be something of the following form.  The terms in Eq. (9) illustrate that if <some property of the distribution is satisfied> then the kernel has desirable property X, rendering it more effective than Y (for some Y).
> > >
> > > In other words, if the explanation provided is in terms of the derived quantities, then the explanation demands those quantities.  And if not, it doesn't, and therefore also doesn't demand the theory.
> > >
> > > The paper has been improved.  But I cannot in good conscious recommend a paper, despite having impressive results, without adequate motivation, so I will keep my score.  I believe that the authors can use the theory to provide insight, and look forward to reading it when they do.

---

> > > > ### Author Response · Authors · 2023-11-23
> > > > **Reply to Reviewer PQpi**
> > > >
> > > > Thank you for your reply.
> > > >
> > > > > I do not see an exact NTK for finite M.
> > > >
> > > > As written in our previous reply to you, while the kernel can be exactly determined, there is no closed-form formulation for it if $M$ is finite. Therefore, there are no specific formulas either. The kernel induced by a finite number of trees asymptotically approaches the closed-form kernel described in Theorem 2 as the number of trees increases.
> > > >
> > > > > For me, the sentence: Such a trend appears to hold true on not only the tic-tac-toe dataset but across a wide range of datasets. Details can be found in the Appendix D.4. is inadequate. Highlighting a particular dataset to make a point is great, being honest about when this point holds, is greater.
> > > >
> > > > The sentence (Such a trend appears…) is not about accuracy comparison with Random Forest but the weight distribution of MKL.
> > > > In Section D.4, we also do not discuss generalization performance but show that the weight distribution of AAI is closer to uniformity compared to the weight distribution of AAA using Figure A.9. In this sense, the claim of this text is correct, and the properties about such a trend of the weight distributions observed in the tic-tac-toe dataset also hold for other datasets.
> > > >
> > > > As you say, it is not always the case that generalization performance improves, which we have also recognized and there is no problem. This is actually mentioned in our Appendix (Section D.5), saying that the generalization performance depends on the data. We are not claiming that performance always improves either. Please note that our paper's claims and such a comparison to Random Forest results are orthogonal.
> > > >
> > > > > What I can infer currently, based on the additional numerical experiments, is that 'greedy' methods perform poorly relative to gradient-based approaches, specifically on datasets in which individual features contain relatively little information, and much more information is in the joint. That totally makes sense. And, that is a point that does not require (or even obviously benefit from) NTK, rather, that point can be made simply by comparing RF to PyTorch Tabular.
> > > >
> > > > Yes, as you say, in our paper, there is no need to use the NTK when discussing generalization performance because it is not our research question.
> > > >
> > > > > The key question for me to motivate this entire line of work (developing NKT for axis-aligned RF) is what insight do we gain specifically from the NKT derivation. I still do not see it.
> > > >
> > > > The motivation is to know better about training of axis-aligned tree ensembles. Our NTK-based theoretical approach enables us to gain insights of tree architectures such as the sufficiency of oblivious trees, that is, any tree architecture can be transformed into a set of oblivious trees with equivalent NTKs (Proposition 2).
> > > >
> > > > > I see why gradient-based methods work better than RF in certain cases. And I see that you can do it, and do neural architecture search, and MKL, etc. And this is all great. I just don't know why. The answer I am looking for would be something of the following form. The terms in Eq. (9) illustrate that if <some property of the distribution is satisfied> then the kernel has desirable property X, rendering it more effective than Y (for some Y). In other words, if the explanation provided is in terms of the derived quantities, then the explanation demands those quantities. And if not, it doesn't, and therefore also doesn't demand the theory.
> > > >
> > > > Combining Theorem 2 and Proposition 1, we prove that any tree architecture can be transformed into a set of oblivious trees with equivalent NTKs (Proposition 2). This is a desirable property, as it can largely prune tree architecture candidates during trial and error. This holds true for any kind of input data, hence it is independent of data distribution. Our theory is necessary to derive these insights.
> > > >
> > > > > The paper has been improved. But I cannot in good conscious recommend a paper, despite having impressive results, without adequate motivation, so I will keep my score.
> > > >
> > > > Again, the motivation is to analyze axis-aligned tree ensembles to gain insights for training. The insights we have gained could not have been acquired without our theory, and we believe it makes sense to use the NTK for theoretical analysis as well. Then our NTK-based approach has successfully derived insights such as those about oblivious trees, which we believe can be a valuable contribution to the ML community.

---

> > ### Comment · Reviewer_PQpi · 2023-11-22
> > **Exact for finite trees?**
> >
> > > Moreover, even in the finite case, the kernel can be precisely derived without any approximation.
> >
> > Thm 1 & 2 both have $M \to \infty$.  I do not see an exact NTK for finite M.  If that exists, and is in this paper, I've missed it. Perhaps I misunderstood the above comment though?

---

### Meta-Review · Area_Chair_v75V · 2023-12-18

**Metareview:**

An interesting paper, which is theoretical in nature but gives useful insights for applications. First I would like to point out that decision trees can be used for heterogeneous variables, e.g. mixing both continuous and categorical variables. It would be interesting to see if the theory can be extended to the latter and more, to the "mixed" broadest setting.

As a side comment, I believe the anslysis, which is restricted to the square loss, does in fact hold for any proper canonical -- and perhaps proper composite -- loss for class-probability estimation, which are the key loss functions to train DTs.

I do consider the argument about faster training (QWBa) a bit dubious: what does faster mean in the paper's context ? For DT induction, it is well-known that the best theory to explain the convergence rates of hard-threshold trees is boosting in Valiant's PAC model, which also applies to any type of variables / split. The argument by the authors that all this "boils down to the oblivious tree reduction" argument is not really a valid argument as the reduction in fact amount to training big equivalent trees with restricted structure. I would thus urge the authors to modulate this argument.

I also agree with the comment from QWBa on the fairness of the comparison with RFs as used in the paper. The authors argue that this is due to the constraint to have the same topology: if we stand by this argument, then RFs cannot be used because indeed they necessitate (lots of) big trees to be accurate. Maybe the authors should have focused on boosting algorithms constrained to the same architecture.

I praise the authors for having added many experiments at rebuttal time *but* they should have put the details of the domains, in particular their size: from what I see, these are small to very small domains by all measures.

Overall, though I value the author's detailed response at rebuttal time and the additional experiments provided, I still remain unconvinced that the paper brings substantial insights compared to what is already known (Kanoh + Sugiyama's ICLR'22 & '23).

**Justification For Why Not Higher Score:**

The paper's contribution remains marginal wrt two previous ICLR papers.

**Justification For Why Not Lower Score:**

N/A

---

### Decision · Program_Chairs · 2024-01-16

Reject